# IRX2 regulates angiotensin II-induced cardiac fibrosis by transcriptionally activating EGR1 in male mice

Zhen-Guo Ma[1,2,3], Yu-Pei Yuan[1,2,3], Di Fan[1,2,3], Xin Zhang[1,2,3], Teng Teng[1,2,3], Peng Song[1,2,3], Chun-Yan Kong[1,2,3], Can Hu[1,2,3], Wen-Ying Wei[1,2,3] & Qi-Zhu Tang [1,2,3] ✉

Cardiac fibrosis is a common feature of chronic heart failure. Iroquois homeobox (IRX) family of transcription factors plays important roles in heart development; however, the role of IRX2 in cardiac fibrosis has not been clarified. Here we report that IRX2 expression is significantly upregulated in the fibrotic hearts. Increased IRX2 expression is mainly derived from cardiac fibroblast (CF) during the angiotensin II (Ang II)-induced fibrotic response. Using two CF-specific *Irx2*-knockout mouse models, we show that deletion of *Irx2* in CFs protect against pathological fibrotic remodelling and improve cardiac function in male mice. In contrast, *Irx2* gain of function in CFs exaggerate fibrotic remodelling. Mechanistically, we find that IRX2 directly binds to the promoter of the early growth response factor 1 (EGR1) and subsequently initiates the transcription of several fibrosis-related genes. Our study provides evidence that IRX2 regulates the EGR1 pathway upon Ang II stimulation and drives cardiac fibrosis.

Sustained cardiac stress results in excessive production of extracellular matrix components, causing cardiac fibrosis. Cardiac fibrosis can increase the stiffness of the left ventricle, disrupt cardiac conduction, and impair systolic and diastolic function[1–3]. The presence of fibrosis has been independently associated with cardiovascular mortality and all-cause mortality[4]. Currently, no evidence-based therapies show significant efficacy against fibrotic diseases, largely because the mechanisms of cardiac fibrosis are unclear. Moreover, our knowledge on the mechanisms of cardiac fibrosis is predominantly derived from experiments performed in cell culture systems or genetic mice with cardiomyocyte-specific manipulation[5].

Upon fibrogenic stimulation, quiescent cardiac fibroblasts (CFs) proliferate and differentiate into myofibroblasts, which express the highly contractile protein α-smooth muscle actin (α-SMA)[6]. Several growth factors and cytokines, such as transforming growth factor-β (TGF-β), angiotensin II (Ang II), platelet-derived growth factor (PDGF), fibroblast growth factor (FGF), and bone morphogenetic protein (BMP), are key driving forces culminating in the cardiac fibrotic

response[7]. These growth factors and cytokines directly bind to their receptors and thus trigger the activation of downstream effector proteins, such as Smad3, to amplify the fibrotic response[7]. Increased knowledge of molecules that regulate multiple pro-fibrotic pathways may be translated into gene-based therapies for the treatment of cardiac fibrosis.

Iroquois homeobox (IRX) genes were first identified in *Drosophila melanogaster*, and this family includes six homeoproteins, named IRX1-IRX6[8]. During neurogenesis, Xiro (the Xenopus analogue of a human IRX member) functions as a repressor of BMP-4 after activation by the Wnt signalling pathway[9]. All six members display specific expression patterns in the heart and play key roles in heart development and function[10,11]. Specifically, IRX4 regulates chamber-specific gene expression in the developing heart, while IRX5 establishes the mouse cardiac ventricular repolarization gradient[10,11]. IRX2 has been shown to be induced by TGF-β[12] and FGF8 during chick hindlimb development[13] and to be regulated by sex-determining region Y box 9 (SOX9)[12], which is a master regulator of cardiac fibrosis[14]. IRX2 was

[1]Department of Cardiology, Renmin Hospital of Wuhan University, 430060 Wuhan, PR China. [2]Cardiovascular Research Institute of Wuhan University, 430060 Wuhan, PR China. [3]Hubei Key Laboratory of Metabolic and Chronic Diseases, 430060 Wuhan, PR China. ✉e-mail: qztang@whu.edu.cn

found to be increased in a mouse model of dilated cardiomyopathy (DCM)[15], and could be activated by mitogen-activated protein kinases[13]. Despite the strong expression of IRX2 in the heart, its effects in diseased hearts, especially in CF activation and fibrotic remodelling, have not been investigated. Here, we identified IRX2 as a CF-enriched transcription factor that promotes the development of pathological cardiac fibrosis by transcriptionally activating growth response factor 1 (EGR1). To the best of our knowledge, this is the first report describing a causative role for IRX2 in fibrotic remodelling.

## Results

### IRX2 expression is upregulated in a murine model of cardiac fibrosis and in human fibrotic hearts

To detect alterations in IRXs during the fibrotic process, isolated mouse CFs were incubated with Ang II, a potent inducer of fibrosis, for 24 h. *Irx2* and *Irx3* expression was increased, while *Irx4* expression was decreased in isolated CFs after Ang II treatment compared to control phosphate-buffered saline (PBS) treatment (Fig. 1A). There were no differences in the other IRX members between the two groups. This finding was confirmed in human CFs, and we observed that *IRX2* and *IRX3* levels were also increased in Ang II-treated human CFs (Fig. 1B). However, Ang II did not decrease *IRX4* expression in human CFs. IRX2 was the IRX family member most consistently and significantly upregulated by Ang II stimulation. Consistent with these data, *Irx2* mRNA expression were significantly increased in Ang II infusion-induced fibrotic hearts, which was indicated by increased mRNA levels of markers of fibrosis, including *Col1* and *α-Sma* (Fig. 1C, Supplementary Fig. 1A). Western blotting results suggested that IRX2 protein expression was increased in the hearts of mice following 12-week-Ang II infusion (Fig. 1D). To enhance the clinical significance of our findings, we detected the expression of IRX2 in human heart samples from patients with DCM. The failing human hearts exhibited fibrosis, as reflected by increased *TGF-β3* and *α-SMA* levels (Supplementary Fig. 1B). As expected, IRX2 expression was significantly upregulated in the failing human hearts compared with control hearts (Fig. 1E). In a

recent report, *Irx2* was also shown to be increased in a mouse model of DCM by RNA-seq[15]. IRX2 was mainly located in the nuclei of CFs (Supplementary Fig. 1C). To identify the cellular source of upregulated IRX2 in fibrotic hearts, we fractionated endothelial cells, cardiomyocytes, and CFs from hearts treated with Ang II for 12 weeks to examine the expression pattern of *Irx2*. *Irx2* mRNA levels were much higher in CFs than in cardiomyocytes and endothelial cells (Fig. 1F). Surprisingly, Ang II infusion had no significant effect on *Irx2* mRNA expression in cardiomyocytes or endothelial cells but markedly increased *Irx2* mRNA levels in CFs (Fig. 1F). In agreement with this finding, immunofluorescence staining for IRX2, Collagen 1 (Col1, a marker for CFs), and cardiac Troponin T (cTnT, a marker for cardiomyocytes) revealed IRX2 expression in Col1-positive fibroblasts but not in cTnT-positive cardiomyocytes (Supplementary Fig. 1D). Next, the IRX2 protein was evaluated in CFs isolated from the hearts of Ang II-infused mice. Compared with the CFs of saline-infused mice, the CFs isolated from mice treated with Ang II infusion showed significantly increased IRX2 protein expression (Fig. 1G). We also found that periostin (POSTN) + /IRX2+ CFs were almost absent in saline or sham-treated hearts, but significantly increased after long-term pressure overload or Ang II infusion (Supplementary Fig. 1E-G).

### IRX2 is a regulator of cardiac fibrosis in mice

To investigate the effect of IRX2 on cardiac fibrosis, we performed 5 independent animal experiments to obtain the most conclusive evidence. First, mice with a conditional knockout allele of *Irx2* (*Irx2*[fl/fl]) were generated and crossed with tamoxifen-inducible *Col1α2*-CreER mice (*Col1α2*-Cre) to obtain conditional fibroblast-specific *Irx2*-deficient mice (*Irx2* cfKO) (Supplementary Fig. 2A). *Irx2* cfKO mice and control littermates were injected intraperitoneally with tamoxifen to induce the depletion of *Irx2* in fibroblasts (Fig. 2A). *Irx2* cfKO mice were viable, fertile and phenotypically normal. After a 2-week washout period, mouse CFs freshly isolated from *Irx2* cfKO mice demonstrated ~61% IRX2 protein loss compared to CFs isolated from *Irx2*[fl/fl] littermates (Fig. 2B). However, the IRX2 protein expression in

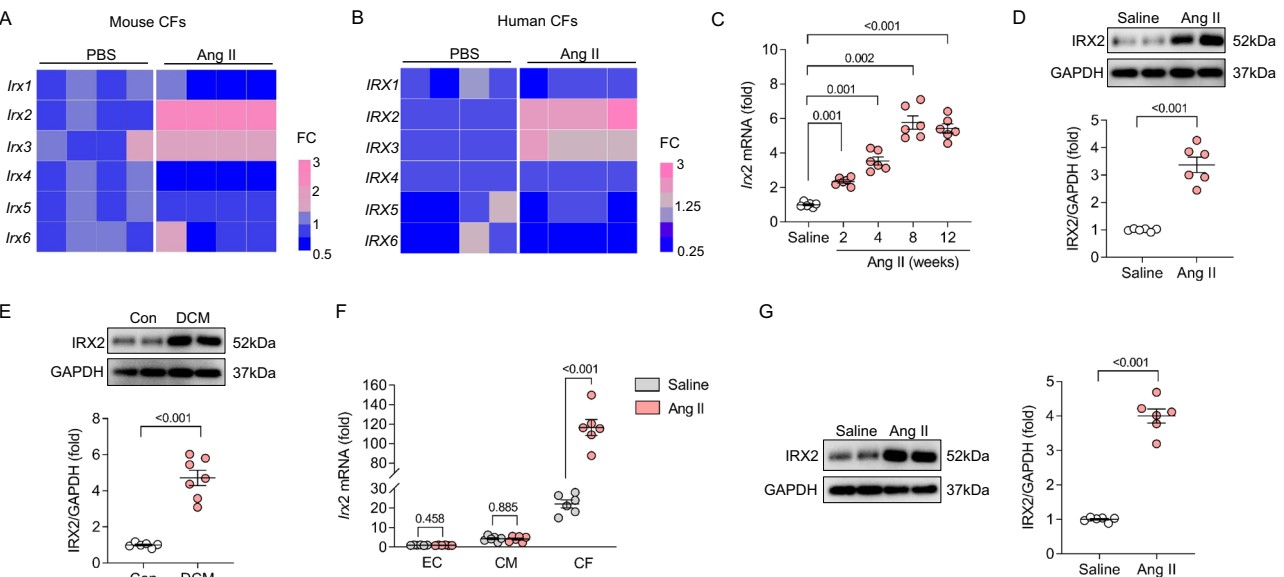

**Fig. 1 | IRX2 expression was increased in fibrotic hearts. A** Relative mRNA levels of 6 *Irx* family members in adult mouse cardiac fibroblasts (CFs) after angiotensin II (Ang II) treatment for 24 h (*n* = 5). **B** Relative mRNA levels of 6 *IRX* family members in human CFs after Ang II treatment for 24 h (*n* = 5). **C** *Irx2* mRNA levels in the heart after Ang II infusion by an osmotic minipump (*n* = 6). **D** Representative western blots and statistical analysis of IRX2 expression in the heart after Ang II infusion for 12 weeks (*n* = 6). **E** Representative western blots and statistical analysis of IRX2 expression in heart samples obtained from patients with dilated cardiomyopathy

(DCM) and control (Con) donors (Con, *n* = 6; DCM, *n* = 7). **F** *Irx2* mRNA levels in endothelial cells (ECs), cardiomyocytes (CMs) and CFs isolated from hearts infused with Ang II for 12 weeks (*n* = 6). **G** Representative western blots and quantification of IRX2 protein expression in CFs isolated from hearts infused with Ang II for 12 weeks (*n* = 6). Data are shown as the mean ± SEM, and analysed using an unpaired two-tailed Student's *t* test. For the analysis in (**C**), one-way ANOVA with Tamhane's T2 test was conducted. Source data are provided as a Source Data file.

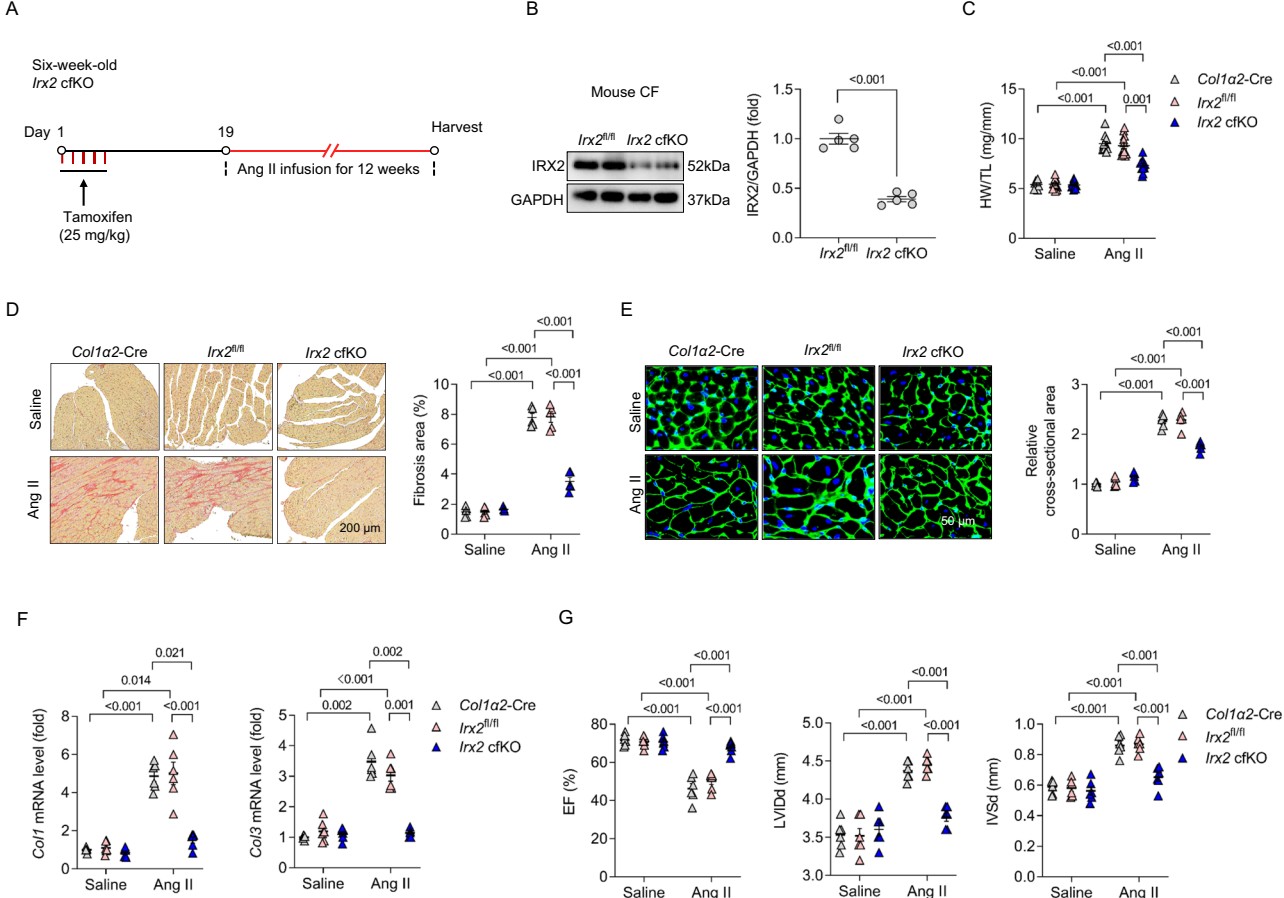

**Fig. 2 | Conditional fibroblast-specific *Irx2*-deficient mice exhibited attenuated fibrotic remodelling in response to angiotensin II (Ang II) infusion.**
**A** Conditional fibroblast-specific *Irx2*-deficient mice (*Irx2* cfKO) were bred by crossing mice with a conditional knockout allele of *Irx2* (*Irx2*fl/fl) with *Col1α2*-Cre mice. *Irx2* cfKO mice and littermate controls were subjected to Ang II infusion for 12 weeks. **B** Representative western blots and statistical analysis of IRX2 protein expression in CFs isolated from *Irx2* cfKO mice and littermate controls (*n* = 5). **C–E** *Irx2* cfKO mice exhibited an attenuated heart weight-to-tibia length (HW/TL) ratio (*n* = 10 mice, *Col1α2*-Cre+Saline; *n* = 12 mice, *Irx2*fl/fl+Saline; *n* = 12 mice, *Irx2* cfKO+Saline; *n* = 12 mice, *Col1α2*-Cre+Ang II; *n* = 11 mice, *Irx2*fl/fl+Ang II; *n* = 12 mice, *Irx2* cfKO+Ang II) (**C**), a reduced fibrosis area (*n* = 5) (**D**), and a decreased cell area of cardiomyocytes (*n* = 5) (**E**) after Ang II infusion. **F** Relative mRNA levels of *Col1* and *Col3* in hearts from *Irx2* cfKO mice and littermate controls after Ang II infusion (*n* = 6). **G** Cardiac function (*n* = 6) was improved in Ang II-infused *Irx2* cfKO mice. Data are shown as the mean ± SEM, and analysed using one-way ANOVA followed by Tukey post hoc test (**E** and **G**) or Tamhane's T2 test (**C**, **D** and **F**). For the analysis in (**B**), an unpaired two-tailed Student's *t* test was conducted. Source data are provided as a Source Data file.

cardiomyocytes was unchanged (Supplementary Fig. 2B). To further examine whether the expression of other *Irx* genes is affected in *Irx2*-deficient CFs, we assessed the mRNA levels of other *Irx* genes in mouse CFs freshly isolated from *Irx2* cfKO mice. No difference in the expression of *Irx1*, *Irx3*, *Irx4*, *Irx5* and *Irx6* could be detected between CFs freshly isolated from *Irx2* cfKO mice and CFs isolated from *Irx2*fl/fl littermates (Supplementary Fig. 2C). To determine the effects of IRX2 deficiency on cardiac fibrosis, *Irx2* cfKO mice and littermate controls were subjected to 12 weeks of Ang II infusion. Upon Ang II infusion, the *Irx2* cfKO mice exhibited a notable alleviation of cardiac enlargement compared with the *Col1α2*-Cre and *Irx2*fl/fl controls, as indicated by decreased heart weight-to-tibia length (HW/TL) ratio (Fig. 2C). *Irx2* cfKO mice exhibited reduced cardiac fibrosis after Ang II infusion compared with littermate controls, whereas no significant fibrosis was observed in the heart of *Irx2* cfKO and littermate controls after saline infusion (Fig. 2D). Ang II-induced cardiomyocyte hypertrophy was moderately attenuated in *Irx2* cfKO mice compared with littermate controls (Fig. 2E). Next, we detected the mRNA levels of two key fibrotic genes and found that *Col1* and *Col3* were upregulated in the heart of *Col1α2*-Cre and *Irx2*fl/fl littermate controls by Ang II infusion and that this induction was largely attenuated in *Irx2* cfKO mice (Fig. 2F). Immunoblotting results then confirmed that the Ang II infusion-

induced protein levels of α-SMA and connective tissue growth factor (CTGF) were blocked by fibroblast-specific depletion of *Irx2* in mice (Supplementary Fig. 3A). The transcriptional levels of hypertrophic markers, including *Anp*, *Bnp* and *β-Mhc*, were significantly suppressed in heart samples from *Irx2* cfKO mice compared with heart samples from littermate controls (Supplementary Fig. 3B). Furthermore, echocardiographic assessment of cardiac function indicated that fibroblast-specific *Irx2* depletion attenuated Ang II-induced cardiac dysfunction, as evidenced by an elevated ejection fraction (EF) and a decreased left ventricular end-diastolic dimension (LVIDd) and inter-ventricular septal thickness at diastole (IVSd) (Fig. 2G, Supplementary Fig. 3C). Ang II infusion induced an increase in systolic blood pressure in *Irx2* cfKO mice and littermate controls; however, there was no significant difference in systolic blood pressure among all three groups (Supplementary Fig. 3D). The heart rate was unchanged in *Irx2* cfKO mice and littermate controls after Ang II infusion for (Supplementary Fig. 3E).

Second, we generated injury-inducible myofibroblast-specific *Irx2*-knockout mice (*Irx2* mfKO). In the healthy adult heart, periostin expression is negligible[16,17]. Upon injury, periostin expression is robustly increased and specifically marks activated myofibroblasts in the fibrotic heart[16,17]. Periostin is not present in resident CFs,

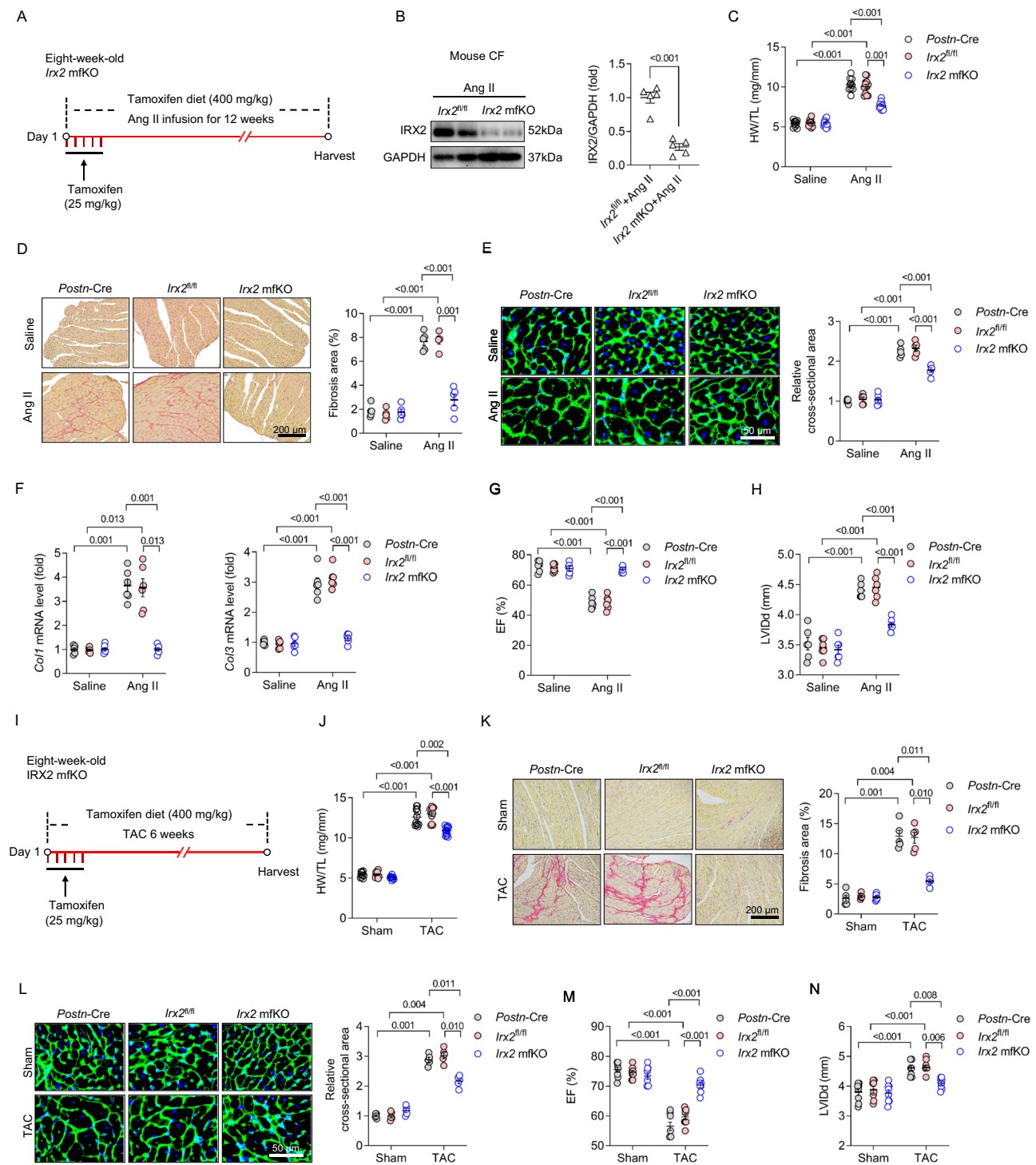

cardiomyocytes, endothelial cells, vascular smooth muscle cells or inflammatory cells[5,16,18] and thus can be used for the specific manipulation of activated myofibroblasts. To generate injury-inducible *Irx2* mfKO mice, *Irx2*fl/fl mice were bred with *Postn*-Cre mice. *Irx2* mfKO mice were also viable, fertile and phenotypically normal. *Irx2* mfKO mice were subjected to 12 weeks of Ang II infusion (Fig. 3A). To investigate the extent of *Postn*-Cre-mediated *Irx2* deletion, we isolated CFs from Ang II-infused *Irx2* mfKO mice and found that the IRX2 protein level was reduced by ~73% in these mice compared to littermate controls (Fig. 3B). However, the *Irx2* mRNA level in cardiomyocytes was unchanged (Supplementary Fig. 4A). Consistent with the findings for *Irx2* cfKO mice, *Irx2* mfKO mice exhibited alleviated fibrotic remodelling upon Ang II infusion, as

indicated by reductions in the HW/TL ratio, fibrosis area, cell area of cardiomyocytes and levels of fibrotic markers (*Col1* and *Col3* mRNA levels), compared with littermate controls (Fig. 3C-F). The increased protein expression of α-SMA and CTGF induced by Ang II infusion was also suppressed by myofibroblast-specific depletion of *Irx2* in mice (Supplementary Fig. 4B). We also found that myofibroblast-specific depletion of *Irx2* decreased the number of POSTN + /IRX2+ cells in response to Ang II infusion (Supplementary Fig. 4C). Further detection of the mRNA levels of hypertrophic markers revealed that myofibroblast-specific depletion of *Irx2* inhibited the pathological elevations in these hypertrophic markers in Ang II-treated mice (Supplementary Fig. 4D). Compared with *Postn*-Cre and *Irx2*fl/fl control mice, *Irx2* mfKO mice showed improved cardiac function

**Fig. 3 | Conditional myofibroblast-specific *Irx2* depletion attenuated angiotensin II (Ang II) or pressure overload-induced cardiac fibrosis in mice.**
**A** Conditional myofibroblast-specific *Irx2*-deficient mice (*Irx2* mfKO) were bred by crossing mice with a conditional knockout allele of *Irx2* (*Irx2*^fl/fl) with *Postn*-Cre mice. *Irx2* mfKO mice and littermate controls were subjected to Ang II infusion for 12 weeks. **B** Representative western blots and statistical analysis of IRX2 protein expression in CFs isolated from *Irx2* mfKO mice and littermate controls after Ang II infusion for 12 weeks (*n* = 5). **C** Heart weight-to-tibia length (HW/TL) ratio (*n* = 10 mice, *Postn*-Cre+Saline; *n* = 11 mice, *Irx2*^fl/fl+Saline; *n* = 11 mice, *Irx2* mfKO+Saline; *n* = 10 mice, *Postn*-Cre+Ang II; *n* = 12 mice, *Irx2*^fl/fl+Ang II; *n* = 12 mice, *Irx2* mfKO+Ang II). **D** Cardiac fibrosis was determined by picrosirius red staining for *Irx2* mfKO mice after Ang II infusion for 12 weeks (*n* = 5). **E** The cell area of cardiomyocytes was determined with wheat germ agglutinin (WGA) staining for *Irx2* mfKO and littermate controls after Ang II infusion for 12 weeks (*n* = 5). **F** The relative mRNA levels of *Col1* and *Col3* were detected in hearts from *Irx2* mfKO mice and control mice (*n* = 6). **G, H** The ejection fraction (EF) (**G**) and left ventricular end-diastolic dimension (LVIDd) (**H**) were detected in these groups (*n* = 6). **I** *Irx2* mfKO mice and littermate controls were subjected to transverse aortic constriction (TAC) surgery for 6 weeks. **J** HW/TL ratio (*n* = 10 mice, *Postn*-Cre+Sham; *n* = 11 mice, *Irx2*^fl/fl+Sham; *n* = 11 mice, *Irx2* mfKO+Sham; *n* = 10 mice, *Postn*-Cre+TAC; *n* = 12 mice, *Irx2*^fl/fl + TAC; *n* = 12 mice, *Irx2* mfKO+TAC). **K** Histological staining showing cardiac fibrosis after TAC surgery in *Irx2* mfKO mice (*n* = 5). **L** Cell area of cardiomyocytes determined with WGA staining after TAC surgery (*n* = 5). **M, N** EF (**M**) and LVIDd (**N**) detected in these groups (*n* = 8). The data were compared by one-way ANOVA. Data are shown as the mean ± SEM, and analysed using one-way ANOVA followed by Tukey post hoc test (**D–H** and **L–N**) or Tamhane's T2 test (**C**, and **J–K**). For the analysis in (**B**), an unpaired two-tailed Student's *t* test was conducted. Source data are provided as a Source Data file.

after Ang II infusion, as indicated by an elevated EF and a reduced LVIDd (Fig. 3G, H).

Third, in addition to Ang II infusion, pressure overload is another well-studied mouse model of cardiac fibrosis[19]. To exclude the possibility that the pro-fibrotic effect of IRX2 is specific to Ang II-induced cardiac fibrosis, we next investigated whether myofibroblast-specific depletion of *Irx2* could attenuate pressure overload-induced fibrotic remodelling in vivo (Fig. 3I). Mice subjected to pressure overload developed cardiac hypertrophy and fibrosis, as indicated by increases in the HW/TL ratio, cardiomyocyte cell area and fibrosis area (Fig. 3J-L). *Irx2* mfKO mice exhibited moderate declines in the HW/TL ratio and cardiomyocyte cell area but a robust decline in the fibrosis area in response to pressure overload (Fig. 3J-L). The increased EF and decreased LVIDd indicated attenuated cardiac dysfunction in *Irx2* mfKO mice compared with control littermates in response to pressure overload (Fig. 3M, N).

Fourth, we generated cardiomyocyte-specific *Irx2*-knockout mice (*Irx2* cmKO) to exclude the effect of cardiomyocyte-derived IRX2 on fibrotic remodelling (Supplementary Fig. 5A). *Irx2* cmKO mice were generated by crossing *Irx2*^fl/fl mice with *α-Mhc*-Cre mice. Mouse cardiomyocytes freshly isolated from *Irx2* cmKO demonstrated ~55% IRX2 protein loss compared to cardiomyocytes isolated from IRX2^fl/fl littermates (Supplementary Fig. 5A). However, the *Irx2* mRNA in CFs was unchanged (Supplementary Fig. 5B). We found that Ang II infusion for 12 weeks induced an increase in the HW/TL ratio, cell area of cardiomyocytes and fibrosis area; however, these pathological alterations were not attenuated by cardiomyocyte-specific depletion of *Irx2* (Supplementary Fig. 5C-F). Cardiomyocyte-specific depletion of *Irx2* did not attenuate Ang II-induced cardiac dysfunction, as indicated by the similar EF among the three Ang II-infused groups (Supplementary Fig. 5G).

Finally, we pursued a gain-of-function approach. We further investigated whether overexpressing *Irx2* in myofibroblasts promoted Ang II-induced cardiac fibrosis in vivo by generating myofibroblast-specific *Irx2* transgenic mice (*Irx2* mfTg). To achieve this, conditional transgenic founder mice expressing the *Irx2* gene (*Irx2* Tg^fl/fl) were bred with *Postn*-Cre mice. This experimental design sketch was showed in Fig. 4A. To prevent potential contamination caused by transgenic leakiness, we evaluated IRX2 protein expression in several organs of *Irx2* mfTg mice and *Irx2* Tg^fl/fl littermates after Ang II infusion. In response to Ang II infusion, IRX2 was increased only in the heart of *Irx2* mfTg mice (Supplementary Fig. 6A), confirming the integrity and specificity of this overexpression system. We also found that IRX2 protein expression was significantly increased in myofibroblasts freshly isolated from Ang II-infused *Irx2* mfTg mice compared with myofibroblasts isolated from *Irx2* Tg^fl/fl littermates (Fig. 4B). However, IRX2 protein expression in cardiomyocytes was unchanged (Supplementary Fig. 6B). As expected, overexpression of *Irx2* in myofibroblasts moderately increased the HW/TL ratio and cell area of cardiomyocytes but robustly

increased the fibrosis area in Ang II-infused mice (Fig. 4C-E). Accordingly, the mRNA levels of *Col1* and *Col3*, as well as the protein expression of α-SMA, were higher in *Irx2* mfTg mice than in control littermates upon Ang II infusion (Fig. 4F, Supplementary Fig. 6C-D). We also found that myofibroblast-specific overexpression of *Irx2* increased the number of POSTN + /IRX2+ myofibroblasts in response to Ang II infusion (Supplementary Fig. 6E). Ang II infusion for 4 weeks could induce cardiac hypertrophy and fibrosis without altering the EF[20,21]. Consistent with these reports, mice infused with Ang II for 4 weeks exhibited an unaltered EF and LVIDd. Mice with conditional myofibroblast-specific overexpression of *Irx2* had a decreased EF and an increased LVIDd after 4 weeks of Ang II infusion (Fig. 4G, H). Taken together, these results suggested that CF-derived IRX2, not cardiomyocytes, contributed to Ang II-induced fibrotic remodelling in mice.

## IRX2 is required for cardiac fibroblast-to-myofibroblast conversion

Next, we investigated the direct effects of IRX2 deletion on the phenotype of CFs. Adult CFs were isolated from *Irx2*^fl/fl mouse hearts, and *Irx2* depletion was achieved by adenovirus-mediated Cre expression. Cre infection led to an ~87% reduction in IRX2 protein expression in vitro (Fig. 5A). *Irx2* depletion significantly decreased *α-Sma* expression in the presence of Ang II (Fig. 5B). Immunofluorescence analysis also revealed that the *Irx2* depletion mediated by Cre expression decreased the α-SMA fluorescence intensity in Ang II-treated CFs (Fig. 5C). We also found that conditioned medium from CFs isolated from *Irx2* cfKO mice significantly attenuated Ang II-induced cardiomyocyte hypertrophy in vitro (Supplementary Fig. 7A). In addition to the Cre-mediated deletion approach, we used an shRNA-mediated interference approach. Two small hairpin RNAs (shRNAs) targeting *Irx2* (sh*Irx2* #1 and #2) significantly decreased IRX2 protein expression in wild-type (WT) mouse CFs (Supplementary Fig. 7B). This shRNA-mediated knockdown of *Irx2* significantly suppressed the upregulation of α-SMA+ cells in response to Ang II treatment (Fig. 5D). In addition, two shRNAs against *Irx2* (#1 and #2) decreased Ang II-induced *Col1* production in CFs (Fig. 5E). Next, we evaluated whether *Irx2* deficiency affects CF proliferation. IRX2 did not affect CF proliferation upon 5% fetal bovine serum (FBS) or Ang II administration (Supplementary Fig. 7C, D). In line with the above findings, we found that *Irx2* overexpression aggravated the Ang II-induced transdifferentiation of fibroblasts into myofibroblasts, as suggested by the increased α-SMA expression in CFs isolated from WT mice (Fig. 5F, Supplementary Fig. 7E). A gain-of-function experiment was also performed with human CFs. Consistent with the findings for mouse CFs, *IRX2* overexpression promoted the Ang II-induced transdifferentiation of fibroblasts into myofibroblasts, which was reflected by increased mRNA levels of *Col1*, *α-Sma* and *Postn* (Supplementary Fig. 7F, G). Next, we performed gene profiling of *Irx2*-overexpressing mouse CFs and control CFs after Ang II

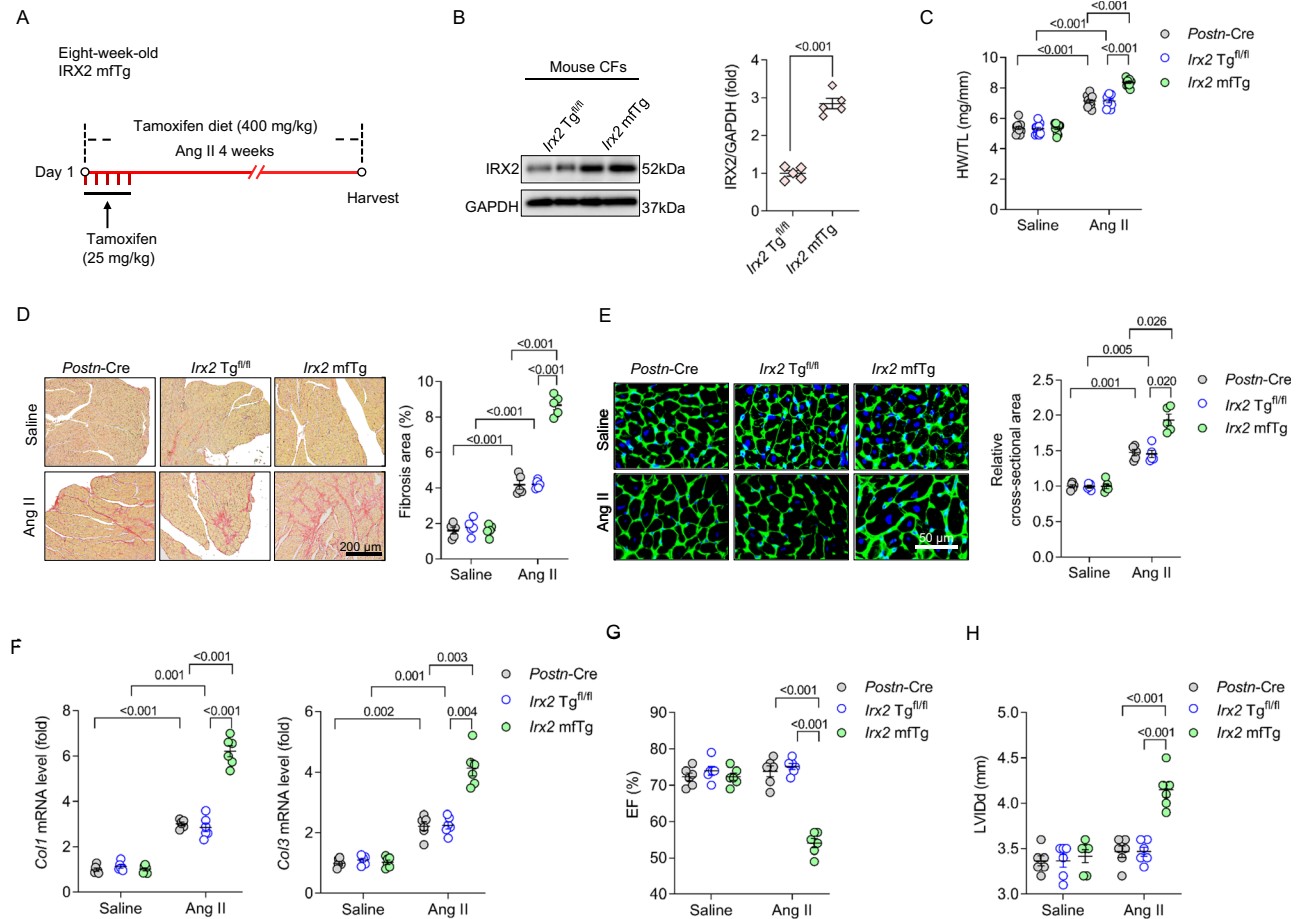

**Fig. 4 | Conditional myofibroblast-specific *Irx2* overexpression promoted angiotensin II (Ang II)-induced fibrotic remodelling in mice. A** Conditional myofibroblast-specific *Irx2*-overexpressing mice (*Irx2* mfTg) were bred by crossing conditional transgenic mice expressing the *Irx2* gene (*Irx2* Tg^fl/fl) with *Postn*-Cre mice. *Irx2* mfTg mice and their littermate controls were subjected to Ang II infusion for 4 weeks. **B** Representative western blots and statistical analysis of IRX2 protein expression in CFs isolated from *Irx2* mfTg mice and littermate controls after Ang II infusion for 4 weeks (*n* = 5). **C** Heart weight to tibia length (HW/TL) ratio (*n* = 12 mice for all six groups). **D** Cardiac fibrosis was determined by picrosirius red staining in *Irx2* mfTg mice after Ang II infusion (*n* = 5). **E** Cell area of cardiomyocytes was

determined with wheat germ agglutinin (WGA) staining in *Irx2* mfTg and littermate controls after Ang II infusion (*n* = 5). **F** Relative mRNA levels of *Col1* and *Col3* were detected in the hearts of *Irx2* mfTg mice and their controls (*n* = 6). **G, H** Ejection fraction (EF) (**G**) and left ventricular end-diastolic dimension (LVIDd) (**H**) were detected in these groups (*n* = 6). The data were compared by one-way ANOVA. Data are shown as the mean ± SEM, and analysed using one-way ANOVA followed by Tukey post hoc test (**C**–**D** and **G**–**H**) or Tamhane's T2 test (**E**–**F**). For the analysis in (**B**), an unpaired two-tailed Student's *t* test was conducted. Source data are provided as a Source Data file.

administration using RNA-seq assays. Under the conditions of fold change >1.1 and adjusted *P* value (padj) <0.01, we found that 2431 genes were upregulated and 2199 genes were downregulated in *Irx2*-overexpressing CFs compared with control CFs after Ang II treatment (Fig. 5G). The heatmap results revealed that the expression of fibrosis-related genes was significantly upregulated in *Irx2*-over-expressing CFs compared with control CFs (Fig. 5H). Some of these upregulated genes were key mediators of the Ang II-induced fibrotic response, including *Tgf-β3, Pdgf-a, Pdgf-b, Smad3, Col1a1, Tipm3* and several *Fgf* family genes (Fig. 5H, Supplementary Table 2). Gene Ontology (GO) term analysis of differentially expressed genes (DEGs) revealed that several signalling pathways were closely related to fibrotic remodelling, specifically cadherin binding, cell adhesion molecule binding, cell–cell junction, focal adhesion, adherens junction, anchoring junction and cell growth (Supplementary Fig. 8A). Reactome term analysis also identified several pro-fibrotic classes, such as assembly of collagen fibrils, extracellular matrix organization, TGF-β receptor signalling, PDGF signalling pathway, vascular endothelial growth factor (VEGF) signalling pathway, ephrin B-mediated forward signalling, Wnt ligand biogenesis and trafficking, and cell–cell communication (Supplementary Fig. 8B).

**EGR1 was identified as a direct downstream target gene of IRX2**
To explore how IRX2 regulates fibrotic remodelling, we identified the downstream targets of IRX2 by chromatin immunoprecipitation-sequencing (ChIP-seq) with two anti-IRX2 antibodies (IRX2-1B7 and IRX2-1C1). Human CFs were infected with an adenovirus carrying *IRX2* and then subjected to Ang II stimulation. Overexpression of *IRX2* was confirmed by western blotting (Supplementary Fig. 7F). A similar spatial distribution of *IRX2* peaks was identified for each of the two IRX2 antibodies (Fig. 6A). ChIP-seq using IRX2-1B7 revealed that the peaks distributed in the promoter transcription site comprised 25.59% of the total peaks, while ChIP-seq using IRX2-1C1 revealed that the peaks distributed in the promoter site comprised 31.41% of the total peaks (Supplementary Fig. 9A, B). Then, a bioinformatic intersection analysis between the upregulated DEGs identified by RNA-sequencing (RNA-seq) and enriched genes revealed to be within the promoter transcription site by ChIP-seq was performed, and 70 genes were identified (Fig. 6B, Supplementary Table 3). We selected the upregulated DEGs in the RNA-seq data because IRX2 acted as a transcriptional activator after activation by mitogen-activated protein kinases[13]. Next, we excluded genes with low abundance in the heart and molecules that have never been related to cardiovascular diseases. After that, the

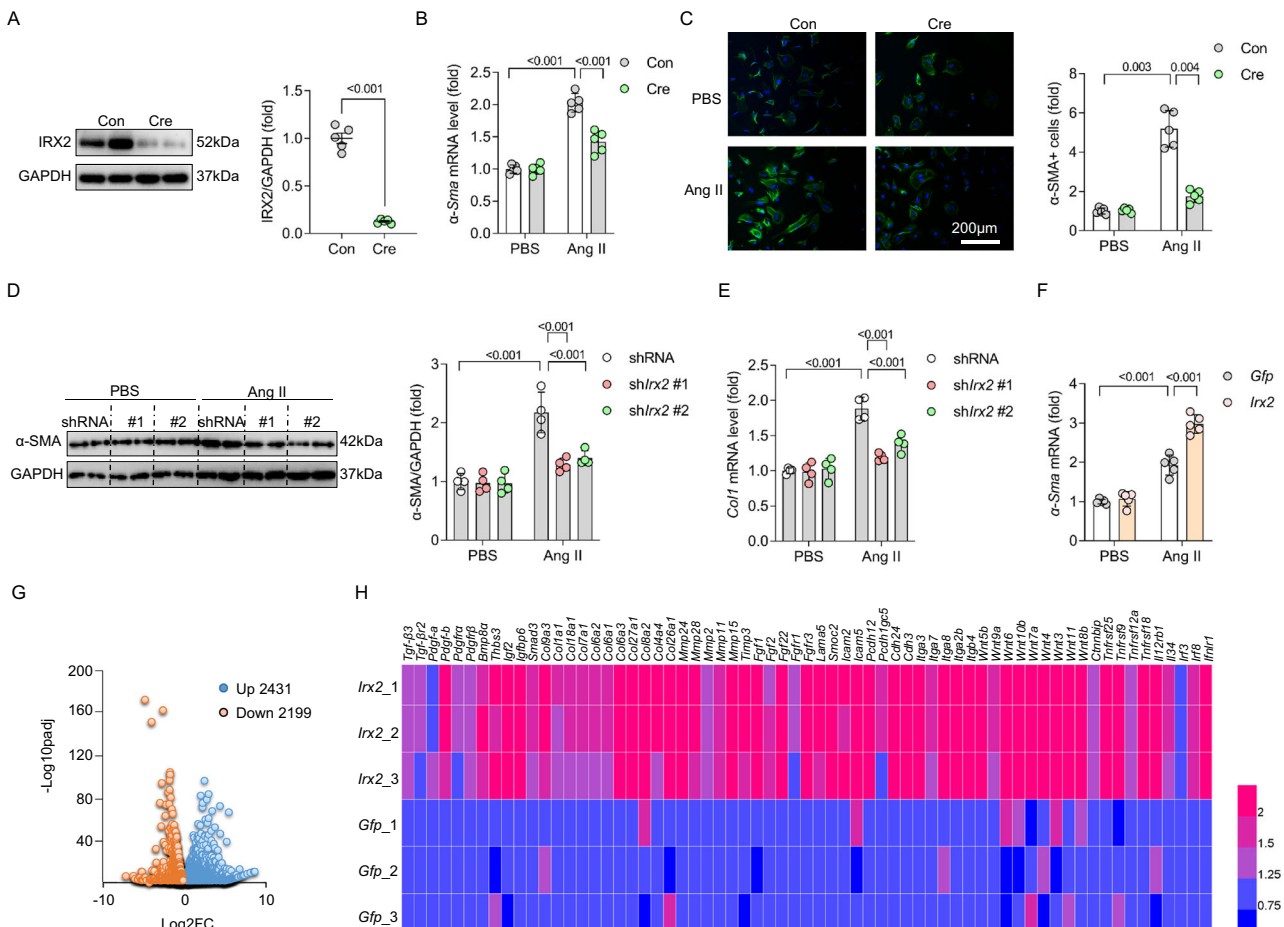

**Fig. 5 | IRX2 is essential for fibroblast to myofibroblast transformation. A** Adult cardiac fibroblasts (CFs) were isolated from *Irx2*^fl/fl mice and infected with an adenoviral vector carrying Cre to deplete *Irx2*. After 24 h of transfection, IRX2 protein expression was detected in CFs (*n* = 5). **B** Relative mRNA level of *α-Sma* in *Irx2*-depleted adult mouse CFs (*n* = 5). **C** Representative images and quantification of α-SMA (green) in Cre-infected *Irx2*^fl/fl CFs in response to Ang II incubation for 24 h. Nuclei are stained with DAPI (blue). (*n* = 5, for each group, 50-60 fields were counted). **D** CFs were isolated from wild-type mice and then transfected with shRNAs targeting *Irx2*. After 24 h of transfection, the cells were subjected to Ang II treatment for 24 h to induce a fibrotic phenotype. After that, the cells were harvested for the detection of α-SMA with western blotting (*n* = 4). **E** Relative mRNA level of *Col1* in IRX2-deficient CFs after Ang II incubation for 24 h (*n* = 4). **F** CFs were isolated from wild-type mice and then overexpressed *Irx2* with an adenovirus

vector, after which they were subjected to Ang II treatment for 24 h. The relative mRNA level of *α-Sma* was detected to reflect fibroblast to myofibroblast transformation (*n* = 5). **G** CFs were isolated from wild-type mice and then overexpressed *Irx2* with an adenovirus vector, after which they were subjected to Ang II treatment for 24 h. Total RNA was extracted, and RNA sequencing was performed. Volcano plots indicating the differentially expressed genes (DEGs) in *Irx2*-overexpressing CFs relative to controls after Ang II administration for 24 h. **H** Heatmap of several differentially expressed genes in *Irx2*-overexpressing CFs relative to controls after Ang II infusion. Data are shown as the mean ± SEM, and analysed using one-way ANOVA followed by Tukey post hoc test (**B** and **D–F**) or Tamhane's T2 test (**C**). For the analysis in (**A**), an unpaired two-tailed Student's *t* test was conducted. Source data are provided as a Source Data file.

number of IRX2-enriched genes was narrowed to 10. The genes were *Gadd45β, Paxillin, Vegfα, Egr1, Nectin2, Serpine2, Klf10, Dusp5, Mapkapk3* and *Col6a2*. Five of them (*Gadd45β, Gadd45β, Nectin2, Klf10* and *Dusp5*) have been reported to protect against fibrotic remodelling[22–26]; thus, they were not the molecules responsible for the pro-fibrotic effects of IRX2. We focused our interest on EGR1 for the reason that EGR1 is a key transcription factor in fibrotic diseases[27]. Increased EGR1 expression were observed in several animal models of fibrosis and human fibrotic diseases[27]. *Egr1*-deficient mice were protected from bleomycin-induced skin and lung fibrosis[28]. More importantly, several EGR1-regulated signalling pathways have been revealed by Reactome term analysis of RNA-seq data, such as assembly of collagen fibrils, extracellular matrix organization, TGF-β receptor signalling, PDGF signalling pathway, VEGF signalling pathway and Wnt signalling pathway[27,29]. IRX2 acted as a transcriptional factor by specifically binding to the ACAnnTGT motif of target genes[30,31]. When searching this motif (ACAnnTGT) among the ChIP-seq peak distributed in the promoter site of *Egr1*, we found that there was two IRX2 binding sites

(Fig. 6C). Next, we performed a luciferase assay with mouse CFs. *Irx2* overexpression dramatically increased luciferase activity in CFs in a dose-dependent manner, suggesting that IRX2 directly stimulated the transcription of *Egr1* (Fig. 6D). More important, after these IRX2 binding sites were mutated, IRX2-mediated upregulation of luciferase activity was abrogated (Fig. 6E). To corroborate this finding, ChIP-PCR was performed with Ang II-treated mouse CFs, and the results showed enrichment of the *Egr1* promoter region in the IRX2 precipitate (Fig. 6F). These results suggested that EGR1 was a direct transcriptional target of IRX2. Consistent with these in vitro findings, *Egr1* mRNA and protein expression were decreased in CFs isolated from *Irx2* cfKO mice in comparison with those isolated from *Col1α2*-Cre and *Irx2*^fl/fl littermate controls after Ang II infusion (Fig. 6G-H). Using a Cre-mediated deletion approach, we also found that *Irx2* deficiency inhibited the pathological elevation in EGR1 in Ang II-treated *Irx2*^fl/fl CFs (Fig. 6I). Conversely, CFs isolated from Ang II-infused *Irx2* mfTg mice exhibited significantly elevated expression of EGR1 compared with CFs isolated from Ang II-infused *Postn*-Cre and *Irx2* Tg^fl/fl mice (Fig. 6J, K). Using

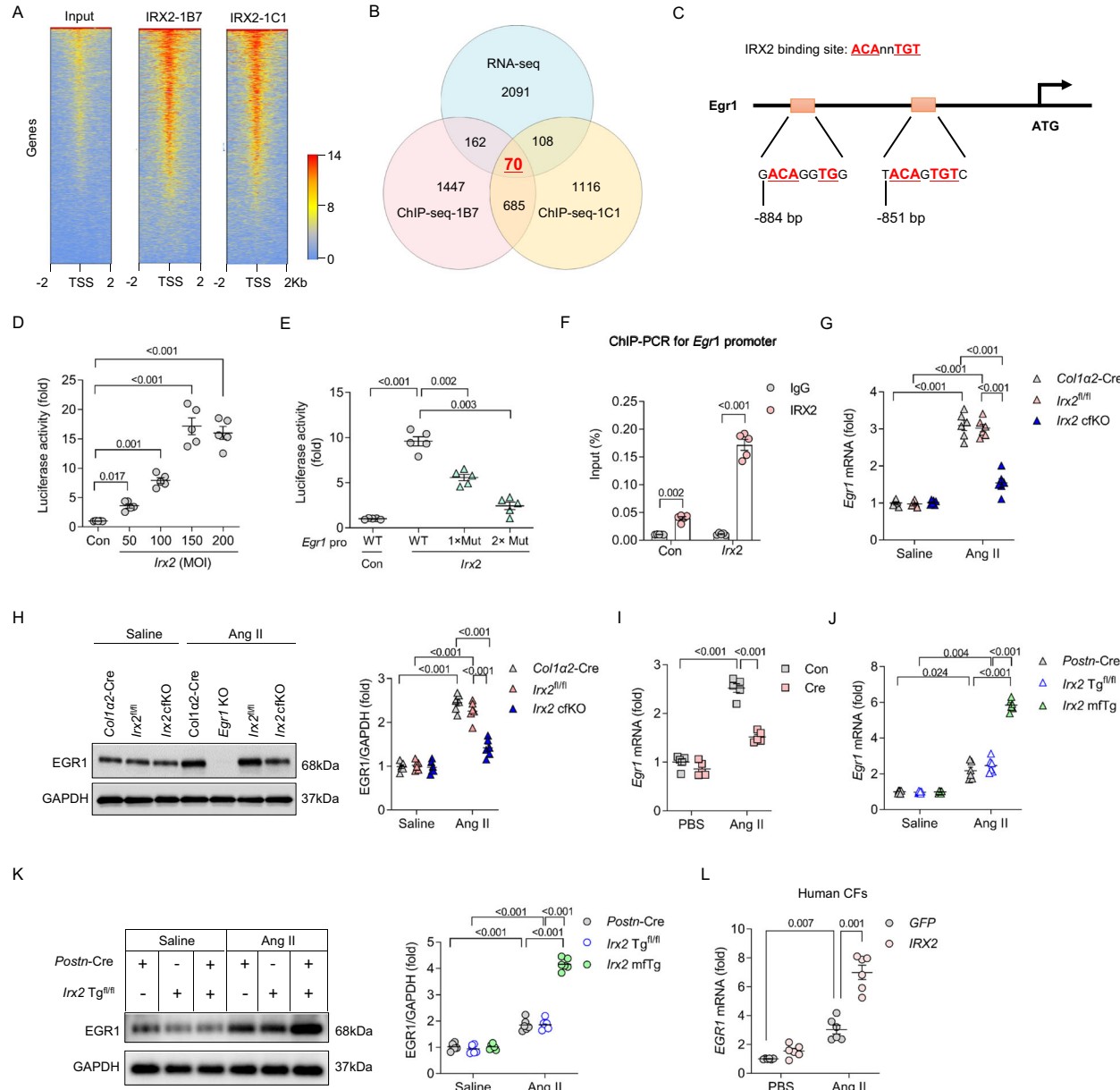

**Fig. 6 | EGR1 was identified as a direct downstream target gene of IRX2 in cardiac fibroblasts that regulates the angiotensin II (Ang II)-induced fibrotic response. A** CFs were infected with an adenovirus carrying *IRX2* and then subjected to Ang II stimulation for 24 h. ChIP-seq was performed with two anti-IRX2 antibodies (IRX2-1B7 and IRX2-1C1). Similar spatial distributions of IRX2 peaks were identified by the two anti-IRX2 antibodies. **B** IRX2-regulated genes identified in the ChIP-seq data (bottom) were overlaid with upregulated differentially expressed genes revealed by RNA-seq (top). With this strategy, 70 genes were identified. **C** Two IRX2 binding sites were identified in the ChIP-seq peak distributed in the promoter site of *Egr1*. **D** The *Egr1* promoter was stimulated by IRX2 overexpression. *Egr1* promoter activity was measured by a luciferase assay upon *Irx2* overexpression in CFs isolated from wild-type mice (*n* = 5 for each group). **E** The mutant (Mut)

promoters (pro) deleting IRX2 binding sites cannot be stimulated by *Irx2* over-expression (*n* = 5 for each group). **F** Independent ChIP-PCR was performed with *Irx2*-overexpressing CFs to verify IRX2 binding to the promoter of *Egr1* (*n* = 5). **G**–**H** *Egr1* mRNA and EGR1 protein expression in CFs isolated from *Irx2* cfKO mice and littermate controls with or without Ang II infusion for 12 weeks (*n* = 6). **I** *Egr1* mRNA levels were detected in CFs with *Irx2* deficiency caused by Cre expression (*n* = 5). **J**–**K** *Egr1* mRNA and EGR1 protein expression in CFs isolated from *Irx2* mfTg mice and littermate controls with or without Ang II infusion for 4 weeks (*n* = 6). **L** *Egr1* mRNA level in *Irx2*-overexpressing human CFs after Ang II treatment for 24 h (*n* = 6). Data are shown as the mean ± SEM, and analysed using one-way ANOVA followed by Tukey post hoc test (**H**, **I** and **K**) or Tamhane's T2 test (**D**–**G**, **J** and **L**). Source data are provided as a Source Data file.

human CFs, we also confirmed that *IRX2* overexpression increased *EGR1* mRNA expression in Ang II-treated CFs (Fig. 6L).

## IRX2 regulated cardiac fibrosis via an EGR1-dependent mechanism

To determine the requirement for EGR1 in the IRX2-mediated fibrotic response, CFs were isolated from *Irx2* Tg^fl/fl^ mice and infected with an adenovirus carrying Cre to overexpress IRX2. After that, EGR1

expression was knocked down. *Egr1* deficiency abolished the IRX2-promoted transdifferentiation of fibroblasts into myofibroblasts in vitro (Supplementary Fig. 10A, Fig. 7A). Ang II-induced collagen accumulation, as reflected by *Col1* and *Col3* mRNA levels, was enhanced by the overexpression of *Irx2*, and this effect of *Irx2* was blocked by *Egr1* knockdown (Fig. 7B, C). To further confirm the role of EGR1 in the IRX2-mediated fibrotic response, we isolated CFs from *Egr1* global knockout mice. These cells were then infected with an

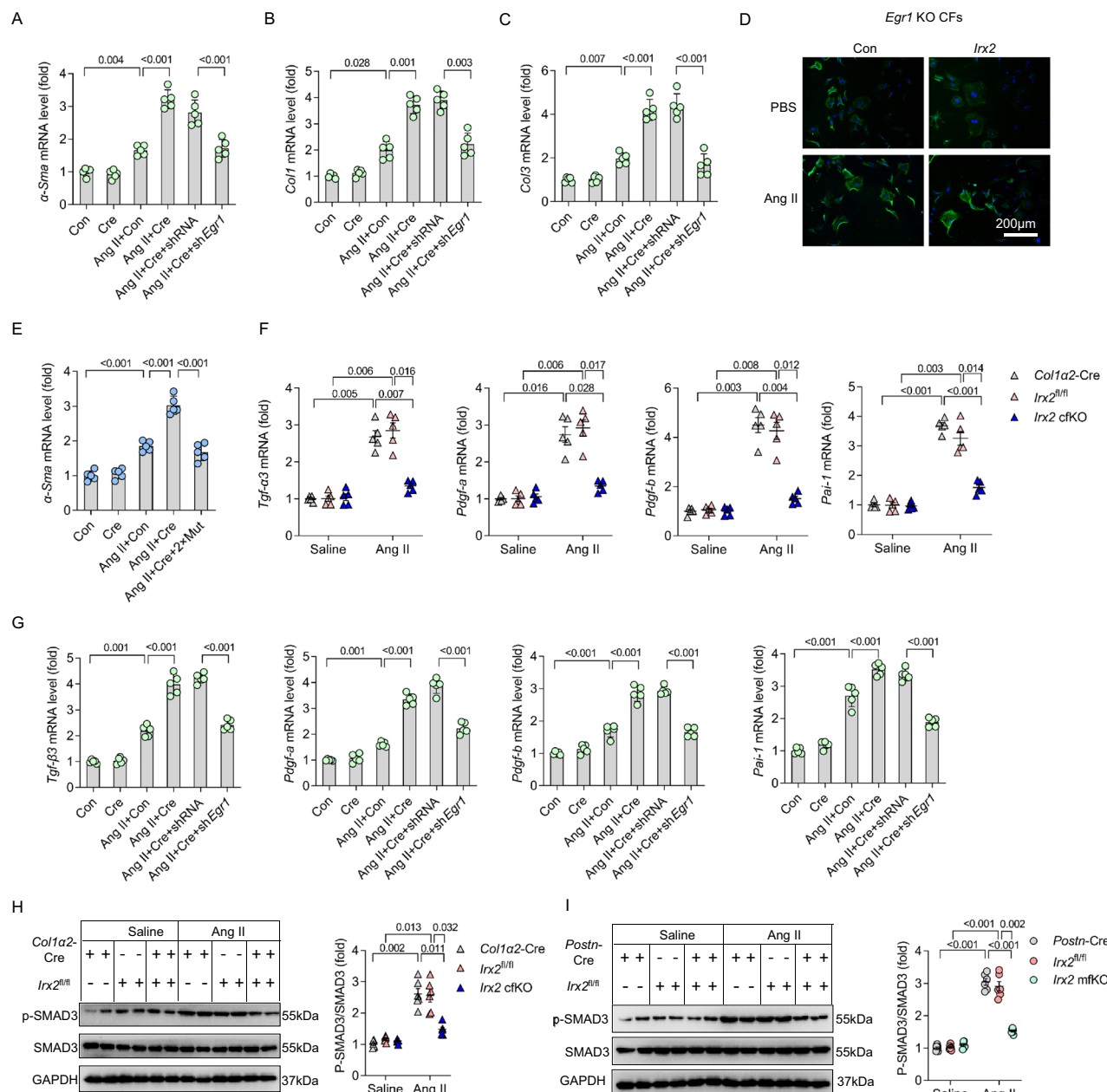

**Fig. 7 | EGR1 was responsible for the IRX2-promoted fibrotic response in angiotensin II (Ang II)-treated cardiac fibroblasts. A–C** Cardiac fibroblasts (CFs) were isolated from *Irx2* Tg$^{fl/fl}$ mice and infected with an adenovirus carrying Cre to overexpress *Irx2*. After that, *Egr1* expression was knocked down, and the CFs were subjected to Ang II treatment for 24 h. After that, the cells were harvested to detect the mRNA levels of *α-Sma* (**A**), *Col1* (**B**) and *Col3* (**C**) (*n* = 5). **D** Representative images of α-SMA (green) in CFs isolated from *Egr1*-knockout mice after Ang II treatment for 24 h. Nuclei were stained with DAPI (blue) (*n* = 5, for each experiment, 50-60 fields were counted). **E** Relative mRNA levels of *α-Sma* in the indicated groups (*n* = 5). **F** Relative mRNA levels of fibrosis-related genes in CFs isolated from *Irx2* cfKO mice

and littermate controls with or without Ang II infusion for 12 weeks (*n* = 5). **G** CFs were isolated from *Irx2* Tg$^{fl/fl}$ mice and infected with an adenovirus carrying Cre to overexpress *Irx2*. After that, *Egr1* expression was knocked down, and the CFs were subjected to Ang II treatment for 24 h. After that, the cells were harvested to detect the mRNA levels of fibrosis-related genes (*n* = 5). **H, I** Western blot analysis was performed to analyse the phosphorylation of Smad3 in CFs isolated from *Irx2* cfKO or *Irx2* mfKO mice and corresponding littermate controls with or without Ang II infusion for 12 weeks (*n* = 6). Data are shown as the mean ± SEM, and analysed using one-way ANOVA followed by Tukey post hoc test (**A**, **C**, **E** and **G**) or Tamhane's T2 test (**B**, **F**, **H** and **I**). Source data are provided as a Source Data file.

adenovirus carrying *Irx2*. EGR1 protein expression was absent in CFs isolated from *Egr1*-deficient mice (Supplementary Fig. 10B). The results demonstrated that IRX2 did not enhance the Ang II-induced transdifferentiation of fibroblasts into myofibroblasts in *Egr1*-deficient CFs (Fig. 7D, Supplementary Fig. 10C). Consistent with the above findings, after these IRX2 binding sites were mutated, IRX2-mediated effect was abrogated, as reflected by *α-Sma* mRNA level (Fig. 7E). To establish a causative relationship between EGR1 and fibrotic remodelling, we performed an *Egr1* gain-of-function

experiment with isolated CFs. *Egr1* overexpression in CFs resulted in spontaneous myofibroblast transdifferentiation (Supplementary Fig. 10D). EGR1 could bind to the promoters of *Tgf-β3*, *Pdgf-a*, *Pdgf-b*, *Col1α2*, *Pai-1* and *Timp3* and increase the expression of these downstream genes[27]. We also found that *Egr1* overexpression significantly increased the mRNA levels of *Tgf-β3*, *Pdgf-a*, *Pdgf-b*, *Col1α2*, *Pai-1* and *Timp3* (Supplementary Fig. 10E). As expected, *Irx2* depletion significantly decreased the Ang II-induced pathological increases in TGF-β3, PDGF-A, PDGF-B, and PAI-1 in CFs (Fig. 7F). Conversely, the

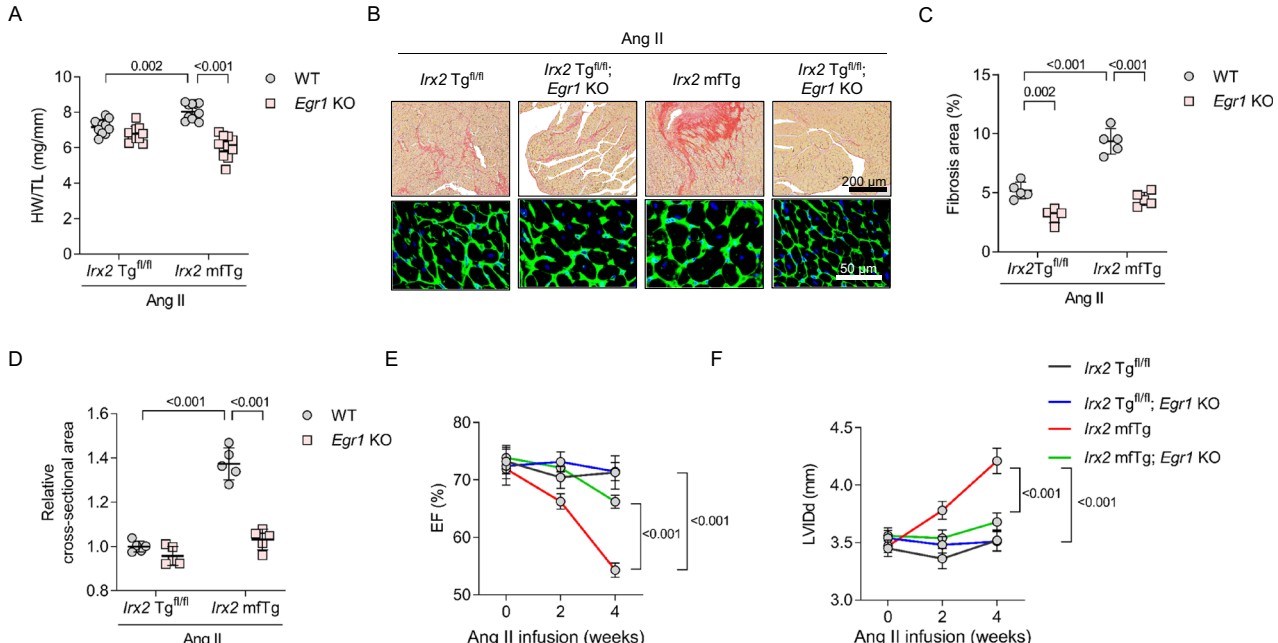

**Fig. 8 | Genetic depletion of *Egr1* attenuated fibrotic remodelling in mice with conditional myofibroblast-specific *Irx2* overexpression after angiotensin II (Ang II) infusion.** Conditional myofibroblast-specific *Irx2*-overexpressing mice (*Irx2* mfTg) were bred with *Egr1* global knockout mice or wild type (WT). The resulting mouse line and littermate controls were subjected to Ang II infusion for 4 weeks. **A** Heart weight-to-tibia length (HW/TL) ratio (*n* = 10 mice, *Irx2* Tg^fl/fl + WT+Ang II; *n* = 10 mice, *Irx2* Tg^fl/fl+*Egr1* KO+Ang II; *n* = 11 mice, *Irx2* mfTg+WT+Ang II; *n* = 11 mice, *Irx2* mfTg+*Egr1* KO+Ang II). **B** Histological staining showed cardiomyocyte

enlargement and cardiac fibrosis after Ang II infusion (*n* = 5). **C** Cardiac fibrosis was determined by picrosirius red staining after Ang II infusion (*n* = 5). **D** The cell area of cardiomyocytes was determined with wheat germ agglutinin (WGA) staining after Ang II infusion (*n* = 5). **E, F** The ejection fraction (EF) (**E**) and left ventricular end-diastolic dimension (LVIDd) (**F**) were detected in these groups (*n* = 6). Data are shown as the mean ± SEM, and analyzed using one-way ANOVA followed by Tukey post hoc test (**A**, **C**, and **D**). For the analysis in (**E**, **F**), repeated measures ANOVA was conducted. Source data are provided as a Source Data file.

Ang II-induced pathological elevations in *Tgf-β3*, *Pdgf-a*, *Pdgf-b*, and *Pai-1* were enhanced by overexpression of *Irx2* in mouse CFs, and these effects could be blocked by *Egr1* knockdown (Fig. 7G). TGF-β3, PDGF and PAI-1 are known to activate Smad-3, which is likely to be a key driving factor in fibrosis[7,32]. Next, we determined the phosphorylation of Smad3 in isolated CFs. As hypothesized, *Irx2* deficiency led to a significant reduction in the phosphorylation of Smad3 in CFs isolated from *Irx2* cfKO or *Irx2* mfKO mice after Ang II infusion (Fig. 7H, I).

**Genetic depletion of *Egr1* attenuated Ang II-induced cardiac fibrosis and improved cardiac function in *Irx2* mfTg mice**

Finally, we confirmed the mechanisms underlying the observed pathological phenotype in *Irx2* mfTg mice, with our hypothesis being that upregulation of EGR1 and its downstream pro-fibrotic genes in IRX2 mfTg mice is the cause, rather than a consequence, of this pathological fibrotic phenotype. To verify this, we evaluated whether genetic depletion of *Egr1* could rescue the observed pathological phenotype in IRX2 mfTg mice after Ang II infusion. IRX2 mfTg mice or IRX2 Tg^fl/fl littermates were bred with *Egr1*-deficient mice. To our surprise, genetic depletion of *Egr1* largely abolished the pathological phenotype caused by conditional myofibroblast-specific overexpression of IRX2, as evidenced by decreases in the HW/TL ratio, cardiomyocyte cell area and fibrosis area (Fig. 8A-D). Genetic depletion of *Egr1* blunted the Ang II-induced increase in fibrosis markers more in the heart of IRX2 mfTg mice than in IRX2 Tg^fl/fl mice (Supplementary Fig. 11A-E). Genetic depletion of *Egr1* also partly restored ventricular function and chamber dimensions (Fig. 8E, F). Together, these findings provide strong evidence that EGR1 upregulation is responsible for the pathological phenotype caused by conditional myofibroblast-specific overexpression of IRX2 in mice.

## Discussion

Currently, although several pharmacological agents have been shown to be effective in heart failure, clinical strategies specifically for the treatment of cardiac fibrosis are not available. Identifying the molecular mechanisms of cardiac fibrosis will pave the way for developing new strategies for the treatment of cardiac remodelling. Herein, we report a detrimental role for IRX2 in the development of fibrosis. Specifically, fibroblast-specific *Irx2* deletion attenuated two different heart fibrotic diseases and improved cardiac function. Myofibroblast-specific *Irx2* gain of function promoted progressive cardiac fibrosis induced by Ang II and accelerated the deterioration of cardiac function in mice. Mechanistic studies demonstrated that IRX2 could directly bind to the promoter region of the *Egr1* gene and subsequently activate downstream pro-fibrotic genes. Our study highlighted the possibility of developing therapeutic strategies targeting IRX2 to attenuate cardiac fibrosis.

Irrespective of the initial cause, injury evoked a sustained fibrotic response in the heart that resulted in a distorted heart architecture and pump dysfunction. CFs are the largest cell population in the heart[33]; however, CFs are still considered to play a secondary role in chronic heart failure. Due to the lack of inducible Cre-expressing lines that are effective for manipulating gene expression in mouse CFs, the regulatory mechanisms of pathological stress-induced fibrotic remodelling have not been fully elucidated. Several studies have revealed the central role of CFs in pathological fibrotic remodelling using the inducible *Postn*-Cre model[5,34,35]. As sources and targets of pathological stimuli, CFs coordinate mechanical and electrical signals among the cell populations of the heart[36]. Growth factors secreted by activated myofibroblasts directly induce cardiomyocyte hypertrophy in a paracrine-dependent manner[37]. Considering the critical role of CFs in cardiac fibrosis, identifying the key mediator of cardiac fibrosis would

be of great clinical significance for designing novel therapeutics to prevent fibrotic remodelling.

Recent studies have revealed that the IRX family of cardiac transcription factors is important in the normal development of the heart in rodents and humans[8,38]. Specifically, loss of IRX4, a member highly expressed in cardiomyocytes, causes cardiomyopathy characterized by cardiac hypertrophy and impaired contractile function[39]. Here, starting from the identification of fibrosis-related IRX members using rodent and human CFs, we found that IRX2 expression was upregulated in murine fibrotic hearts and failing human hearts. This pattern of upregulation was supported by results obtained with a mouse DCM model but challenged by a study reporting that IRX2 expression during the hypertrophic response was unchanged[40]. These incompatible results may be explained by the different phases and degrees of the fibrotic response between these reports. Moreover, we identified IRX2 as a CF-enriched transcription factor, implying a distinct role for IRX2 during the development of cardiac fibrosis.

*Irx2*-deficient mice do not display a notable phenotype in the heart structure or electrical conduction[41]. IRX2 was not essential for normal heart development, likely being functionally redundant with other IRX members. Our results provide the first indication for a role for IRX2 in regulating the pathophysiological process of cardiac fibrosis. In both our Ang II and transverse aortic constriction (TAC) models of fibrosis, we observed a reduction in fibrosis that was accompanied by an improvement in cardiac function in mice with CF- or myofibroblast-specific *Irx2* depletion. Surprisingly, but not unexpectedly, this attenuated fibrotic response was not observed in mice with cardiomyocyte-specific *Irx2* depletion, suggesting that IRX2 derived from CFs, not cardiomyocytes, plays a critical role in Ang II-induced fibrotic remodelling in mice. Most importantly, *Irx2* mfTg mice demonstrated a pathological phenotype and deteriorated cardiac function upon Ang II infusion. Inconsistent with our findings, Wang and his colleagues observed that lentivirus injection-induced *Irx2* knockdown prevented mice from Ang II-induced cardiac hypertrophy in mouse heart[42]. These incompatible results may be explained by the different manipulation of myocardial IRX2. Several studies have raised concerns that Col1α2-driven gene deletion was of limited value in deciphering the mechanism of cardiac fibrosis[43,44]. The use of Col1α2-driven *Irx2* deletion didn't compromise our main finding that IRX2 is a master regulator of pathological cardiac fibrosis. The limitation of Col1α2-driven IRX2 deletion was addressed with the use of *Postn*-Cre-mediated IRX2 deletion.

Upon stimulation, CFs rapidly differentiate into myofibroblasts, which express the highly contractile protein α-SMA[45,46]. This conversion is a signature of an ongoing fibrotic response. Consistent with the fibrosis phenotype observed in vivo, fibroblast-to-myofibroblast transformation was significantly blocked after *Irx2* loss of function but enhanced by *Irx2* gain of function. Proliferation of activated CFs is another hallmark of fibrosis and plays an important role in the pro-fibrotic effects induced by Ang II[3]. In our study, IRX2 did not affect Ang II-induced CF proliferation, which was consistent with a previous report demonstrating that IRX2 promoted leukaemia cell differentiation without affecting cell proliferation[47]. IRX2 gain of function didn't impact fibrotic phenotype basally, but deteriorated Ang II infusion-induced fibrotic phenotype, implying that there was a post-translational upstream switch for activating IRX2 transcription activation. Consistent with this hypothesis, Matsumoto et al reported that mitogen-activated protein kinase kinases 1, which was continually activated during fibrotic process, phosphorylated IRX2 protein to induce its activation in chick[13]. Deciphering the upstream regulation IRX2 would be of great translational significance. The limitation of our study is that whether IRX2 inhibition can reverse fibroblast-to-myofibroblast transformation remains unclear. This warrants further investigation.

Identification of the fibrosis-promoting role of IRX2 raised another important question: what factors are responsible for IRX2-mediated cardiac fibrosis? There have been no reports describing the downstream targets of IRX2. We performed a bioinformatic intersection analysis between transcriptomic and ChIP-seq data and identified *Egr1* as an IRX2 target. Our data showed that IRX2 could directly bind to the promoter of the *Egr1* gene and promote its transcription in CFs. To the best of our knowledge, this is the first report describing the regulation of *Egr1* by IRX2. Interestingly, Ang II-induced EGR1 expression was blocked by *Irx2* depletion but enhanced by *Irx2* gain of function. Given the critical role played by EGR1 in fibrosis, EGR1 modulation might be a key mechanism by which IRX2 promotes fibrotic remodelling in mice. This hypothesis was verified by our studies assessing genetic depletion of *Egr1*, which rescued the pathological phenotypes observed in mice with myofibroblast-specific IRX2 overexpression. These data were in agreement with a report showing that Egr-1 deficiency protected against organ fibrosis in mice[48]. However, there sounds a different voice that global *Egr1* knockout increased the number of fibrotic loci[49]. *Egr1* depletion exacerbated, rather than attenuated, carbon tetrachloride-induced hepatic fibrosis[50]. These incompatible results suggested EGR1 played a context-dependent role in fibrogenesis, possibly determined by the injury onset and activated cell types.

Accumulating evidence has demonstrated that the transcription factor EGR1 acts as an important mediator of fibroblast activation and the fibrotic response[29]. Herein, we also found that increased EGR1 expression directly induced fibroblast-to-myofibroblast transformation and regulated several fibrosis-related genes. The EGR1-dependent regulation of these fibrosis-related genes has been reported previously[29,51]. In our study, IRX2 regulated several pro-fibrotic factors in Ang II-infused CFs in an EGR1-dependent manner. The upregulation of TGF-β3 and PDGF members, as well as the activation of the Smad3 signalling pathway, contributed to the pathological phenotype observed in mice with myofibroblast-specific *Irx2* overexpression.

In our study, we also found that myofibroblast-specific *Irx2* depletion, but not cardiomyocyte-specific *Irx2* depletion, could attenuate cardiac hypertrophy, and improve cardiac systolic function. Growth factors and cytokines produced by CFs, were closely involved into the development of cardiac hypertrophy via paracrine-mediated cell-cell communications[37]. The decreased secretion of myofibroblasts-derived pro-hypertrophic factors (TGF-β, PDGF-A and PDGF-B) in mice with myofibroblast-specific *Irx2* depletion in response to Ang II infusion might explain the improvement in cardiac hypertrophy and systolic function.

In summary, this study identified IRX2 as a critical regulator of cardiac fibrosis via the transcriptional activation of EGR1. These observations advance our understanding of the mechanisms of pathological cardiac fibrosis.

## Methods

### Study Design
All animal studies were conducted in accordance with the guidelines for the Care and Use of Laboratory Animals published by the United States National Institutes of Health (NIH Publication, revised 2011) and the guidelines of the Animal Care and Use Committee of Renmin Hospital of Wuhan University. The objective of this study was to investigate the role of IRX2 in pathological cardiac fibrosis. The sample size of the animals was determined according to our previous study[52]. Investigators were blind to the animal experimental group. No samples were excluded from the analysis. Figure legends list the sample size, number of biological replicates, number of independent experiments.

### Animals and treatments
Mice with a conditional knockout allele of *Irx2* (*Irx2*^fl/fl) were acquired from Biocytogen Pharmaceuticals (Beijing, China). To achieve this

knockout, the second and third exons were flanked by loxP sites, and thus, two single guide RNAs (sgRNA4 and sgRNA11) targeting the second and third exons were constructed. Purified Cas9 mRNA and two sgRNAs (sgRNA4: 5′-TGCCAAGACCCGCTCTGGCGGGG-3′; sgRNA11: 5′-GAAGAAAGACTACACCCACCAGG-3′) were then injected into zygotes, which were subsequently implanted into pseudopregnant mice. From 174 transferred zygotes, we identified one founder (#24) that possessed floxed second and third exons within the same allele. Successful generation of this conditional knockout allele was validated by touchdown PCR with the KOD-FX enzyme (#KFX-101, TOYOBO, Osaka, Japan) using the following *Irx2*-specific primers: forward, 5′-GCATCAAGCTTGGTACCGATCAGAGCTCGTAGGCTGATTTCC CT-3′; and reverse, 5′-ACTTAATCGTGGAGGATGATGCAGTTAGTGAGG ACAGGCTACAGC-3′. Tamoxifen-inducible *Col1α2*-CreER mice (strain #029567) and injury-inducible *Postn*-Cre mice (Postn^tm2.1(cre/Esr1*)Jmol, strain #029645) were purchased from The Jackson Laboratory (Bar Harbor, USA). *Irx2* cfKO mice were bred by crossing *Irx2*^fl/fl mice with *Col1α2*-Cre mice. To induce Cre recombinase expression in *Irx2* cfKO mice, adult mice were intraperitoneally injected with tamoxifen (25 mg/kg per day, #T-5648, Sigma–Aldrich, St. Louis, USA) for five consecutive days. The dose of tamoxifen was selected according to our previous study[19]. After a 2-week washout period, IRX2 in isolated CFs was detected by western blotting. Injury-inducible conditional myofibroblast-specific *Irx2* mfKO were bred by crossing the obtained *Irx2*^fl/fl mice with *Postn*-Cre mice. These mice were intraperitoneally injected with tamoxifen (25 mg/kg per day) for five consecutive days beginning from the first day of Ang II infusion and fed with a diet containing 400 mg/kg tamoxifen citrate (#130860, Envigo) from the first day of Ang II infusion until the time of sacrifice. Mice carrying the *α-Mhc*-MerCreMer construct (A1cf^Tg(Myh6-cre/Esr1*)1Jmk, strain #005657) were obtained from The Jackson Laboratory; these mice were used in our previous studies[19]. Cardiomyocyte-specific *Irx2* cmKO mice were generated by mating *Irx2*^fl/fl mice with *α-Mhc*-Cre mice. To further confirm the role of IRX2 in cardiac fibrosis, we used myofibroblast-specific *Irx2* mfTg mice. Conditional transgenic founder mice expressing the *Irx2* gene (*Irx2* Tg^fl/fl) were generated by Cyagen Biosciences Inc. (Guangzhou, China). The scheme for the construction of these mice is shown in Supplementary Fig. 6A. Successful generation of these conditional transgenic mice was validated by PCR with *Irx2*-specific primers (forward, 5′-GATCCCCATCAAGCTGATCCG-3′; reverse, 5′-TACGGGTGGTAGCTGATGGC-3′) and internal control PCR primers (forward, 5′-GCAGAAGAGGACAGATACATTCAT-3′; reverse 5′-CCTACTGAAGAATCTATCCCACAG-3′). Injury-inducible *Irx2* mfTg mice were bred by crossing *Irx2* Tg^fl/fl mice with *Postn*-Cre mice. These mice were intraperitoneally injected with tamoxifen (25 mg/kg per day) for five consecutive days beginning from the first day of Ang II infusion and fed with a diet containing 400 mg/kg tamoxifen citrate from the first day of Ang II infusion until the time of sacrifice. The transgenic leakiness of *Irx2* mfTg mice was evaluated by evaluating IRX2 protein expression in the liver, lungs, kidneys, and heart. To verify the hypothesis that EGR1 mediates IRX2-induced pro-fibrotic effects, we investigated the role of EGR1 in IRX2-mediated cardiac fibrosis using *Egr1* global knockout mice (strain #012924, The Jackson Laboratory). EGR1 deficiency was confirmed by western blotting. These mice were free access to water and food with five mice per cage. They were kept in a specific pathogen-free (SPF) condition with 20-25 °C temperature and 45-55% humidity on a regular 12 hour light/dark cycle.

All animal experiments were performed in male mice to avoid possible shielding of estrogens. A cardiac fibrosis model was established by long-term Ang II infusion. Briefly, 8- to 10-week-old *Irx2* cfKO or injury-inducible *Irx2* mfKO mice, along with corresponding age-matched littermate control mice, were implanted with osmotic mini-pumps (ALZET; Durect Corp) containing Ang II (1,000 ng/kg/min, H1705, Bachem, Bubendorf, Switzerland) or the same volume of saline

for 12 weeks in our and other studies[19,20]. These osmotic minipumps were implanted subcutaneously in mice under anaesthesia with 2% isoflurane. To test the hypothesis that *Irx2* mfTg mice are more vulnerable to Ang II and exhibit accelerated development of cardiac dysfunction in response to Ang II infusion, *Irx2* mfTg mice and age-matched littermate control mice were subjected to Ang II infusion for 4 weeks because 4 weeks of Ang II infusion can induce cardiac hypertrophy and fibrosis without altering the left ventricular EF[20,21].

Pressure overload is another well-studied mouse model of cardiac fibrosis. Briefly, thesse mice were anesthetized by intraperitoneal injections of 3% pentobarbital sodium, and the adequacy of anaesthesia was assessed by the pedal withdrawal reflex, breathing, and response to operation. The left chest was exposed by performing a ministernotomy, and the thoracic aorta was identified at the second intercostal space. Transverse aortic constriction was preformed with a 27-G needle using a 7-0 silk suture. After that, the needle was removed, and the thoracic cavity was closed. During the surgery, the body temperature was maintained by a warming pad[19,53]. Temgesic (0.1 mg/kg/day) was used to relieve postoperative pain. The adequacy of ligation was confirmed by validating increased blood flow velocity using Doppler echocardiogram. The mice in the sham groups underwent a similar procedure without ligation. Echocardiography was performed at 6 weeks after surgery. Mice were sacrificed by cervical dislocation at the end of these experiments.

## Blood pressure measurement and echocardiography
Twelve weeks after Ang II infusion, the systolic blood pressure of conscious mice was measured by a non-invasive tail-cuff method. Transthoracic echocardiography was performed with a Vevo® 3100 high resolution Preclinical Imaging 496 System (FUJIFILM VisualSonics, Toronto, Canada) equipped with a 30-MHz linear ultrasound transducer[19,54,55], and echocardiographic parameters were averaged from three to four cardiac cycles. During echocardiography, particular attention was given to avoid exerting excessive pressure on the chest of the mice anaesthetized with 2% isoflurane.

## Histological analysis and immunofluorescence staining
Hearts were arrested in 10% KCl and fixed with 10% neutral formalin. After dehydration and embedding in paraffin, cardiac tissues were subsequently sectioned. Alexa Fluor™ 488 conjugate of wheat germ agglutinin (WGA, #W11261, Invitrogen, Carlsbad, USA) was used in order to measure cross-sectional area of cardiomyocytes. Sections were incubated with WGA (1:100) at 37 °C for 2 h. SlowFade™ Gold Antifade Mountant with DAPI (#S36939, Invitrogen) was used to stain nuclei. The value of cross-sectional area from each heart was averaged from more than 30 cells of an individual heart. These sections were then stained with picrosirius red (PSR) to evaluate the volume of fibrosis in the hearts. The value of fibrotic area was averaged from more than 10 fields from an individual heart.

To determine the cellular location of IRX2 in hearts, sections were stained with anti-IRX2 (1:200, #AF0552, Affinity Biosciences, CA, USA), anti-cTnT (1:200, #ab8295, Abcam, Cambridge, UK) or anti-Col1 (1:200, #ab6308, Abcam) overnight, followed by an incubation with an Alexa Fluor 568-conjugated goat anti-rabbit IgG (H + L) cross-adsorbed secondary antibody (#A-11011, Invitrogen) and Alexa Fluor 488-conjugated goat anti-mouse IgG (H + L) cross-adsorbed secondary antibody for 1 h[19]. The specificity of the anti-IRX2 antibody from Affinity Biosciences was verified in conditional *Irx2*-deficient mice or *Irx2*-deficient CFs. To determine POSTN + /IRX2+ CFs, sections were stained with anti-IRX2 (1:200) and anti-POSTN (1:200, # 66491-1-Ig, Proteintech Group, Inc, Chicago, USA) overnight. The number of POSTN + /IRX2+ CFs was determined by investigators blinded to specific treatment. A minimum of 10 fields were analyzed every mouse and the average number of POSTN + /IRX2+ CFs for each sample was then calculated. To determine the subcellular location of IRX2, CFs were stained with

anti-IRX2 at a dilution of 1:100 overnight, followed by an incubation with an Alexa Fluor 488-conjugated goat anti-rabbit IgG (H + L) highly cross-adsorbed secondary antibody (#A-11034, Invitrogen) for 1 h. To assess the expression of α-SMA, CFs were fixed with 4% formaldehyde, permeabilized in 0.1% Triton X-100 and stained with anti-α-SMA (#ab7817, Abcam) at a dilution of 1:200 overnight, followed by an incubation with an Alexa Fluor 488-conjugated goat anti-mouse secondary antibody (#A11017, Invitrogen) for 1 h. SlowFade™ Gold Antifade Mountant with DAPI (#S36939, Invitrogen) was used to stain nuclei. To assess the area of cardiomyocytes in vitro, cardiomyocytes were stained with anti-α-actinin (#05-384, Merck Millipore) at a dilution of 1:200 overnight, followed by an incubation with an Alexa Fluor 568-conjugated goat anti-mouse IgG (H + L) cross-adsorbed secondary antibody (#A-11004, Invitrogen). Slides were examined in a blinded manner, and the results generated with Image-Pro Plus 6.0 (Media Cybernetics, Silver Springs, MD, USA) were confirmed independently by two authors.

## Human heart samples

Human heart samples have been used in previous reports[19,56]. Human fibrotic heart samples were taken from the left ventricle of DCM patients undergoing cardiac transplantation. Control heart samples were obtained from normal heart donors whose hearts were not suitable for transplantation for non-cardiac reasons. Detailed information for the donors and patients with DCM was previously presented[56]. For donor group, 2 females and 4 males were included. For DCM group, 2 females and 5 males were included. All donors and patients with DCM have signed written informed consent, and all human experiments were conducted in accordance with the Declaration of Helsinki and approved by our Institutional Review Board (Renmin Hospital of Wuhan University Review Board).

## Western blotting and quantitative real-time PCR

Total protein was extracted from heart tissues with lysis buffer containing a protease inhibitor cocktail (#11852700, Roche, Basel, Switzerland) and phosphatase inhibitor (#04906837001, Roche). Protein concentrations were determined with a BCA assay kit (#23227, Invitrogen). The proteins were electrophoresed by SDS−PAGE (10% gel)[19,54,57] and transferred to polyvinylidene fluoride membranes (#IPFL00010, Millipore, MA, USA). After blocking with 5% non-fat milk for 2 h, the membranes were incubated with the following primary antibodies overnight at 4 °C: anti-IRX2 (1:500, #AF0552, Affinity Biosciences), anti- CTGF (1:500, #ab6992, Abcam), anti-EGR1 (1:1000, #55117-1-AP, Proteintech, Manchester, UK), anti-Smad3 (1:500, #9513 S, Cell Signaling Technology, Danvers, MA, USA), anti-phosphorylated Smad3 (p-Smad3, 1:500, #8769, Cell Signaling Technology), and anti-GAPDH (1:2000, #2118, Cell Signaling Technology), followed by incubation with horseradish peroxidase-conjugated secondary antibodies at room temperature for 1 h. Blots were visualized with an enhanced chemiluminescence kit (#1705061, Bio−Rad Laboratories, Hercules, USA) and analysed using Image J software (Bio-Rad, Hercules, CA, USA).

Total RNA was extracted from frozen samples with TRIzol reagent (#15596018, Invitrogen). Total mRNA isolated from heart samples was reverse transcribed into complementary DNA (cDNA) with a Transcriptor First Strand cDNA Synthesis Kit (#04896866001, Roche, Basel, Switzerland). Reactions were quantified with LightCycler 480 SYBR Green 1 Master Mix (#04707516001, Roche) and appropriate primers (Supplementary Table 1). mRNA levels were normalized to those of *Gapdh*.

## Cell culture and treatment

Adult mouse CFs were isolated as described previously[5,20,35]. Briefly, adult mice (6 to 8 weeks old) were anaesthetized with 2% isoflurane, and the hearts were excised and mounted on a Langendorff apparatus.

After washing away the blood with Hanks' Balanced Salt Solution (#24020117, Gibco, Grand Island, NY, USA), the hearts were perfused with hyaluronidase and collagenase II (2 mg/mL, #LS004176, Worthington, Lakewood, USA) for 25 min. After digestion, CFs were purified by removing other cells via a differential attachment technique. CFs were then cultured in Dulbecco's modified Eagle's medium (DMEM)/F12 (#51445 C, Gibco) supplemented with 10% FBS for 48 h. Then, the culture medium was replaced with serum-free DMEM/F12 for 12 h to synchronize the cells before experiments. After that, the CFs were treated with Ang II (1 μmol/l) for 24 h to mimic a fibrotic phenotype. To deplete IRX2 proteins, CFs isolated from IRX2[fl/fl] mice were infected with an adenovirus carrying Cre or a control (Con) for 4 h at a multiplicity of infection of 50 and then treated with Ang II for 24 h. To further confirm IRX2 deficiency in the fibrotic phenotype, CFs isolated from wild-type (WT) mice were infected with adenoviral genome particles carrying shRNAs against *Irx2*. The targeting sequences were as follows: 5′-TGGCCATTATCACCAAGATGACC-3′ (sh*Irx2* #1), 5′-GGCT CTGAGTGTAAGGACAAGTTTG-3′ (sh*Irx2* #2), and CATTGCTGGTTA-TACAATCTCTGTT (sh*Irx2* #3). To further assess the role of IRX2 in the fibrotic response, CFs were infected with an adenovirus carrying *Irx2* or *Gfp* for 4 h at a multiplicity of infection of 50. shRNA targeting mouse *Egr1* (sequence: 5′-GCTGCTTCATCGTCTTCCTCT-3′) was used to further clarify the role of EGR1 in the IRX2-mediated fibrotic response in vitro. In addition, CFs were isolated from *Egr1*-knockout mice and WT mice and infected with an adenovirus carrying *Irx2*. Then, the cells were harvested to evaluate the role of EGR1 in the IRX2-mediated fibrotic response. CFs in the first and second passages were used in our studies. All the adenoviral vectors and shRNAs were provided by Hanbio Biotechnology Co., Ltd. (Shanghai, China).

Normal human CFs were provided by Cell Applications Inc. (San Diego, CA). These human CFs were negative for contamination with several viruses, bacteria, and fungi. Human CFs were maintained in DMEM/F12 containing 10% FBS for 48 h and then infected with an adenovirus carrying *IRX2* for 4 h at a multiplicity of infection of 50. To verify the hypothesis that EGR1 is responsible for the fibrotic effects of IRX2, *Egr1* expression in human CFs was knocked down with an shRNA targeting human *EGR1* (sequence: 5′-CCAUGGACAACUACCCUAAT T-3′). After Ang II stimulation for 24 h, the cells were harvested for the detection of several fibrotic markers.

Neonatal rat cardiomyocytes (NRCMs) were isolated and cultured as described previously[56,57]. Briefly, hearts were quickly excised from neonatal Sprague−Dawley rats (1 to 3 days old) under sterile conditions and washed with DMEM/F12 3 times. After that, the atria and aorta were discarded, and the ventricles were minced into 1- to 2-mm pieces and digested with D-Hanks containing 0.125% trypsin (#25200, Gibco) for 20 min (5 min each time, 4 times) at 37 °C. The cells were then centrifuged at 250 g for 5 min, and the pellet was resuspended in DMEM/F12 containing 15% FBS. CFs were separated via a differential attachment technique. After that, NRCMs were plated in 35-mm dishes coated with gelatine. Bromodeoxyuridine (0.1 mmol/l) was used to prevent neonatal fibroblast contamination. To evaluate the crosstalk between CFs and cardiomyocytes, CFs were isolated from *Irx2* cfKO mice and *Irx2*[fl/fl] mice and then stimulated with Ang II for another 24 h to induce a fibrotic phenotype. Then, conditioned medium from these myofibroblasts was collected and used to treat NRCMs in the absence or presence of Ang II for another 24 h. After that, the cell area of cardiomyocytes was detected by immunofluorescence staining. Cardiac endothelial cells were isolated from adult mice using CD31 microbeads (Miltenyi Biotec)[19].

## Analysis of cardiac fibroblast proliferation

CFs isolated from *Irx2*[fl/fl] mice were infected with an adenovirus carrying Cre or a control for 4 h. Twenty-four hours after infection, the CFs were cultured in DMEM/F12 with or without 5% FBS for 3 days. In addition, *Irx2*[fl/fl] CFs were infected with an adenovirus carrying Cre and

thereafter subjected to Ang II treatment for 3 days. CF proliferation was examined by an MTT assay.

## RNA-seq library preparation, sequencing and analysis

CFs were infected with an adenovirus carrying *Irx2* or *Gfp* and then subjected to Ang II stimulation for 24 h. After that, total RNA was extracted, and RNA degradation and potential contamination were determined with 1% agarose gels. RNA purity was determined with a NanoPhotometer® spectrophotometer (IMPLEN, CA, USA). The RNA concentration was measured with a Qubit® RNA Assay Kit on a Qubit® 2.0 Fluorometer (Life Technologies, CA, USA). RNA integrity was detected using an RNA Nano 6000 Assay Kit for the Bioanalyzer 2100 system (Agilent Technologies, CA, USA). Library construction was performed with a NEBNext® UltraTM RNA Library Prep Kit Illumina® (NEB, USA) with 3 µg RNA per sample following the manufacturer's recommendations. Library quality was evaluated with an Agilent Bioanalyzer 2100 system. After cluster generation with TruSeq PE Cluster Kit v3-cBot-HS (Illumina), the library preparations were sequenced on an Illumina HiSeq platform, and paired-end reads were generated. Three independent biological replicate samples were sequenced for each group. After removing low-quality reads with FASTX-Toolkit, the high-quality clean data were aligned to a mouse genome (mm10) with HISAT2. Differentially expressed gene (DEG) analysis between the two groups was conducted with the DESeq2 R package. RNA-seq and subsequent analyses were performed by Novogene (Beijing, China).

## ChIP, ChIP-seq and ChIP-PCR

Human CFs were infected with an adenovirus carrying *IRX2* for 4 h and then subjected to Ang II stimulation for 24 h. After that, the cells were cross-linked with 1% formaldehyde for 10 min. Then, glycine (0.125 mol/l) was added to terminate the cross-linking reaction. The cross-linked cells were harvested with lysis buffer containing a complete protease inhibitor mixture, and the nuclei were collected by centrifugation at 2000 g for 5 min. After that, the nuclei were further treated with nuclear lysis buffer and sonicated (15 times, 30 s on/off). Precipitation was performed with an anti-IRX2 antibody (#PCRP-IRX2-1B7, Developmental Studies Hybridoma Bank, Houston, USA; or #PCRP-IRX2-1C1, Developmental Studies Hybridoma Bank) incubated overnight at 4 °C. The DNA–protein complexes were used for immunoblot analysis to verify the precipitation of IRX2. After confirmation, DNA was purified with a QIAquick PCR purification kit (#28104, Qiagen) after reversing the cross-linking and performing protease K digestion. Sequencing libraries were prepared by using a VAHTS Universal DNA Library Prep Kit for Illumina V3 (#ND607, Vazyme). The library products corresponding to 200-500 bps were enriched, quantified and finally sequenced on a NovaSeq 6000 sequencer (Illumina) with the PE150 model. ChIP-seq with input DNA and ChIP DNA was performed by Seqhealth (Wuhan, China). Raw reads were processed to include only high-quality reads using Trimmomatic. Clean data were compared to a human genome (hg19) using STAR software (version 2.5.3a). Only uniquely aligned reads were retained for subsequent bioinformatic analysis. Read distribution analysis was performed with RSeQC (version 2.6). To identify a significant binding site of IRX2, peak calling was performed with MACS2 (version 2.2.7.1), with the input used as the control. Peaks with $P$ values < 0.01 were kept for further analyses. Peak annotation was performed with BEDTools (Version v2.25.0). Motif analysis was performed with Homer (version 4.10). ChIP-PCR was performed using primers (forward 5′-CCCTGGATGACAGCGATAGAA-3′; reverse 5′-CAAATAAGGGTTGTTCCGGCG-3′) spanning the IRX2 binding site in the *Egr1* promoter.

## Luciferase reporter gene assay

To evaluate the regulation of EGR1 by IRX2, an *Egr1* promoter clone (chr18 + :35019534-35021123) was obtained from iGene Biotechnology Co., Ltd. (#MPRM16005, Guangzhou, China). This promoter clone was inserted into a pGL3 promoter luciferase vector (*Egr1*-Luc). CFs isolated from WT mice were cultured for 48 h and then infected with an adenovirus carrying *Irx2* for 4 h. After infection, the CFs were electrotransfected with *Egr1*-Luc (0.03 µg) with the Neon® Transfection System (pulse voltage: 1700 V, pulse width: 20 ms) and treated with Ang II. After Ang II incubation for 24 h, the cells were harvested, and luciferase activity was measured using dual-luciferase assay kits (Promega, Madison, USA). The *Egr1* promoter 1×mutant or 2×mutant were also generated by one-step PCR mutagenesis to delete one (ACAGGTGG) or two IRX2 binding sites (ACAGGTGG and ACAGTGT). To confirm that IRX2 promoted fibroblast activation directly through interaction with the *Egr1* promoter, *Irx2*-overexpressed CFs were infected with an adenovirus carrying mutated *Egr1* promoter with two IRX2 binding sites deletion.

## Statistical analysis

Results are expressed as the mean ± SEM. We used the Shapiro–Wilk test to evaluate data normality. Comparisons among more than two groups were performed by one-way ANOVA followed by the post hoc Tukey test when the data had a normal distribution and ANOVA found no variance in homogeneity; otherwise, Tamhane's T2 post hoc test was used (SPSS 26.0, SPSS Inc., Chicago, IL). Comparisons between two groups were performed using an unpaired Student's t test. Repeated-measures ANOVA was performed to compare differences among groups over time. No samples were excluded from the analysis. Statistical significance was assigned at $p < 0.05$.

## Reporting summary

Further information on research design is available in the Nature Portfolio Reporting Summary linked to this article.

## Data availability

The data that support the main findings are available within the main text and the supplementary, and Source data. The datasets generated for the RNA-seq are available through the Gene Expression Omnibus (https://www.ncbi.nlm.nih.gov/geo/query/acc.cgi?acc=GSE236454). The datasets generated for the ChIP-seq are available through the Gene Expression Omnibus (https://www.ncbi.nlm.nih.gov/geo/query/acc.cgi?acc=GSE236455). Source data are provided with this paper.

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

## Acknowledgements

This work is supported in part by grants from the Key Project of the National Natural Science Foundation (No. 81530012), the National Natural Science Foundation of China (No. 82070410, 82270248), The Young Top-notch Talent Cultivation Program of Hubei Province, Knowledge Innovation Program of Wuhan-Basic Research, and the Fundamental Research Funds for the Central Universities (2042021kf0205). We thank Dr Yu-Chao Yu (Inheregene Biotechnology Co., Ltd) for bioinformatics assistance.

## Author contributions

Zhen-Guo Ma, Yu-Pei Yuan, and Qi-Zhu Tang conceived the project, designed the experiments, and planned the research. Zhen-Guo Ma, Yu-Pei Yuan, Teng Teng, Xin Zhang, Peng Song performed most of the experiments. Chun-Yan Kong, Can Hu, Wen-Ying Wei, and Di Fan, Zhen-Guo Ma collected data and performed part of the experiments. Zhen-Guo Ma, Yu-Pei Yuan, and Qi-Zhu analyzed the data and interpreted the results. Zhen-Guo Ma, Yu-Pei Yuan, and Qi-Zhu drafted the paper. Di Fan and Peng Song provided technical support and edited the manuscript. All authors provided feedback and approved the final submission.

## Competing interests

The authors declare that they have no competing interest.
