## [Peer Review File · Nature Communications]

REVIEWER COMMENTS

Reviewer #1 (Remarks to the Author):

In this manuscript from Dr. Qi-Zhu Tang's group, the author has 1st time identified the role of IRX2 in fibroblast activation, cardiac fibrosis, and cardiac function. The studies are elegantly done with multiple mouse models; these include 2 FB-specific IRX2 KO, CM-specific IRX2 KO, FB-specific IRX2-TG, and FB-specific IRX2-EGR1 double KO animals. There is an appreciation for employing both the gain-of-function and the loss-of-function models. As this is 1st paper reporting IRX2 role in fibrosis, the novelty is high, and so is the quality of data and presentation. Additional two stress models (Ang-II infusion and TAC) are employed to test the hypothesis; thus, the study's rigor is high. I have the following comments to further strengthen the paper before final publication.

1. The rationale for choosing the IRX family to investigate in the context of cardiac fibroblast activation and fibrosis is not clear. In other words, it's a random selection rather than based on a big data analysis or a screening experiment. Addressing this will significantly improve the rationale/justification.
2. A recent paper, "Int J Cardiol . 2023 Jan 15;371:332-344 (PMID: 36181956) has showed, "IRX2 knockdown prevented mice from Ang II-induced ventricular dysfunction, cardiac hypertrophy, inflammation and fibrosis in mouse heart, and it decreased the levels of cardiac hypertrophy-related markers, oxidative stress response, and apoptosis of Ang II-treated cardiomyocytes." Of note, both papers employ the Ang II infusion as a primary model of cardiac stress. Therefore, this recently published paper needs to be cited, and the finding of the current paper need to be discussed with the reported results. However, this reviewer feels that the present paper is much more advanced with two FB-specific models. Thus, there is no concern about a lack of novelty/overlap, but it needs to be discussed.
3. Col1 α 2-Cre driven FB-targeting is questionable (PMID: 36289614, 34572061, 28436487). However, this concern is somewhat addressed with the Postn-Cre model. Nevertheless, the limitation associated with the Col1 α 2-Cre model should be discussed/acknowledged. This reviewer requests the authors to consider PostnMCM, and TCF21MCM to target residential FBs and activated FBs, respectively. In the FB field, a consensus is developing that these two models are currently the best tool to target cardiac fibroblasts (PMID: 34572061, 27447449).
4. Targeting all fibroblasts with the Col1 α 2-Cre model led to a 61 % deletion; however, the more restrictive deletion strategy with the Postn-Cre model (myofibroblast only) led to 73% deletion. Authors need to explain this and discuss it in the text.
5. In the α MHC-Cre-driven CM-specific model, 55 % deletion is significantly less than the vast literature with this model. Please clarify/discuss.
6. Mouse age, tamoxifen treatment, and deletion blots are crucial; authors need to bring the experimental design sketch, deletion blot, etc., for all the models in the main figures.

7. Authors may want to change the word "foetal" to "fetal" to be consistent with the North American writing style like the rest of the manuscript text.

8. In the text, authors cited fig 2F-H). However, figure 2 and its legend ends at 2G only; there is no H. Please correct.

9. Fig 6H, and Fig S9A, it seems there are two EGR1 bands. The author must authenticate the correct band with the respective control (complete KO control samples).

10. Similarly, in Fig. S4, the IRX2 blot shows two bands. This need to be addressed.

11. Furthermore, authors are requested to submit the uncut western blot images for all figures as an online-only figures.

Reviewer #2 (Remarks to the Author):

In this manuscript, Ma, et al., identify a novel role for the Iroquois homeobox protein 2 (Irx2) in regulating cardiac fibrosis. The authors begin from the observation that the six Irx family proteins show distinct expression patterns in the heart yet their roles in the adult heart remain understudied. They quickly identify Irx2 as being particularly upregulated by AngII treatment to induce fibrotic remodeling in isolated murine and human cardiac fibroblasts. They then show that Irx2 is also upregulated in cardiac tissue from patients with dilated cardiomyopathy, confirming the potential clinical significance of this observation. Using a tamoxifen-induced Col1a2-CreER driver, they knock out Irx2 in cardiac fibroblasts and show that this can rescue or partially rescue phenotypes associated with fibrosis in the murine hearts, specifically, normalized heart weight, picosirius red staining, expression/protein of myofibroblast markers, and echocardiogram. They then knock out Irx2 only in activated cardiac fibroblasts using an injury-induced Postn-Cre and convincingly show that myofibroblast-specific loss of Irx2 is sufficient to rescue AngII- and pressure-induced cardiac fibrosis while myofibroblast-specific overexpression of Irx2 exacerbates AngII-induced cardiac fibrosis. However, the subsequent argument that Irx2 is therefore necessary for fibroblast to myofibroblast activation is less convincing.

The authors then turn to identifying the mechanism of Irx2 regulation of cardiac fibrosis, which is where this manuscript is at its strongest. Using RNA-Seq and CHIP-seq, the authors identify the gene for transcription factor Egr1 as a direct target of transcription activation by Irx2. They then show that in the absence of Egr1, the impact of knockout or overexpression of Irx2 on their metrics of AngII-induced fibrosis is nearly completely mitigated. Thus, Ma, et al., convincingly demonstrate that in response to profibrotic stimulation, Irx2 can regulate the fibrotic activity of cardiac fibroblasts by activating the transcription of the profibrotic transcription factor Egr1. This work, with a few adjustments, will have significant impact on the study of cardiac fibrosis by helping to elucidate part of the mechanism of cardiac fibroblast activation to myofibroblast. It also suggests a novel druggable target in Irx2 for treatment of cardiomyopathies, which if borne out could significantly improve prognosis for these patients. The problems with the manuscript are minor when weighed against its strengths, but the impact of the work will be greatly increased by addressing them. Specific comments below:

1. The rationale for studying the Iroquois homeobox family of proteins in the context of CFs and cardiac fibrosis is lacking. There is an abrupt jump from “these proteins are expressed in the heart, and many have known roles in development” to “we studied whether these proteins impact CFs in cardiac fibrosis.” The rationale for studying *Irx2* particularly in this context, however, is strong scattered throughout the manuscript:

a. *Irx2* is induced by *Fgf8* and TGF- β ;

b. MAPK phosphorylation activates *Irx2* transcription activation and inhibits *Irx2* transcription repression;

c. *Irx2* is regulated by *Sox9*, a key regulator of cardiac fibrosis;

d. *Irx2* (and *Irx1*) were upregulated in a mouse model of DCM;

Moving or copying (b) (lines: 270-1) and (d) (lines: 95-6) to the introduction will start the results on firmer footing.

2. The manuscript and figures do an excellent job of selecting and expounding on *Egr1* as a major mechanism through which *Irx2* can regulate fibrosis, but a brief primer on what is known about *Egr1* promoting fibrosis in the heart and other organs would be beneficial. Additionally, are the global *Egr1* knockout mice protected against AngII-induced fibrosis? Reference (28) of this manuscript (PMID: 21511034), which is used in the rationale for *Egr1* selection cites an article (PMID: 10710348) where global *Egr1* knockout increased the number of fibrotic loci basally and in response to adrenoceptor agonists. If this is also true for the *Egr1* knockout mice used in this study, it does not invalidate the authors conclusions, but it does complicate the picture and should be addressed in the discussion.

3. Because the protein abundance of *Irx2* does not impact fibrotic phenotypes basally, but only after profibrotic stimulus, there must be a post-translational upstream switch for activating *Irx2* transcription activation. The authors heavily hint that this switch is phosphorylation by MAPK induced by AngII or pressure, and I am inclined to accept that. The scope and novelty of this study is in its identification of the downstream mechanism of *Irx2*. However, the question of upstream activation arises immediately given that in Figure 1 AngII increases *Irx2* expression, but increasing *Irx2* expression in Figure 2 is not sufficient to basally increase fibrosis. The simplest way to address this would just be to explore alternative hypotheses for upstream regulation in the discussion, but experiments with CFs expressing phospho-mimetic and phospho-dead *Irx2* mutants or something similar nodding to upstream regulation would add to the study's impact.

4. The authors mutated the putative *Irx2* binding sites in the *Egr1* promoter in a luciferase assay and showed that this reduced *Irx2*-dependent luciferase activity. The most straightforward way to demonstrate that *Irx2* promotes fibroblast activation directly through interactions with the *Egr1* gene would be to either mutate those binding sites in isolated CFs or design a CRISPRd system to competitively inhibit *Irx2* at those specific *Egr1* binding sites and show a reduction in α SMA and/or Postn expression and positive cells and ECM proteins after AngII treatment. As the manuscript stands, it is clear that *Irx2* can induce *Egr1* transcription and that loss of *Egr1* prevents *Irx2*-dependent fibrotic remodeling. While this circumstantially suggests that *Irx2* is regulating fibrosis by binding and activating transcription at the *Egr1* gene, the more direct experiment would show this conclusively, adding to the significance of the current study.

5. In Figure 2G and elsewhere, representative echocardiograms need to be included in the figure or at least in a corresponding supplementary figure.
6. For Figures 5C and 7D (as well as for sections taken from hearts after saline or AngII treatment), it will be meaningful to show the percentage of α SMA and/or Postn positive cells alongside the changes in fluorescence intensity. A global reduction in just the α SMA fluorescence intensity is not adding anything to the immunoblot showing the same thing and westerns are better indicators of protein abundance than IF micrographs. However, if there is a significant reduction in the number of α SMA (and especially Postn given it is the Cre driver) positive cells in the AngII treated *Irx2* knockout cells, then that is highly suggestive of a defect in fibroblast to myofibroblast conversion rather than a defect in myofibroblast fibrotic activity.
7. Is there a rationale for using the *Col1a2*-CreER mouse over the better characterized *Tcf21*-CreER mouse? The authors include adequate controls to address this concern, and none of the conclusions hinge on the *Col1a2*-cre mouse anyway because the authors immediately shift to the more physiologically relevant injury-inducible model.
8. Figure 6F could be made clearer by adding to the boxplot itself that it is a ChIP-PCR for the *Egr1* promoter. The legend and text state this clearly, but having the plot clear that it is measuring *Egr1* will be helpful to someone skimming the figures. Taking a broader view, it would help to clarify for each occurrence (ideally in the figure itself but at least in the legend) what each color is for all the IF micrographs.
9. Figure S7 would be clearer if the color legend for the *p*_{adj} values were removed for both A and B, adding a note in the legend that only highly significant pathways (whatever the cutoff is, be it *p*_{adj}<0.01 or just 30 and 20 most significant, respectively) are shown.

Reviewer #3 (Remarks to the Author):

Cardiac fibrosis is a common pathology in heart disease, but its molecular mechanisms of regulatory processes remain unclear. The authors hypothesized that *IRX2*, an *IRX* family gene involved in cardiac development and induced by Ang II administration, is a regulator of cardiac fibrosis, and investigated the role and mechanism in cardiac fibrosis using gain- and loss-of-function mouse models. Mechanistically, the authors found *Egr1* as a direct target gene of *IRX2*. *IRX2* induced by AngII activates *Egr1* to promote cardiac fibrosis, and AngII-induced cardiac fibrosis is suppressed in *Egr1* knockout mice. This manuscript elegantly demonstrates the role of *Irx2* in cardiac fibrosis. However, to argue that *Irx2* is the "master regulator of cardiac fibrosis", it would be desirable to clarify whether there is a same fibrotic mechanism mediated by the *IRX2*-*Egr1* pathway in a more physiological model of cardiac fibrosis, myocardial infarction. Furthermore, recent single cell analyses suggested that cardiac fibroblasts in heart failure undergo a dynamic transition from a steady state to a pathological state. I believe it is important to clarify whether *IRX2* regulates the transition from steady state to myofibroblasts or whether it regulates the pro-fibrotic properties of myofibroblasts.

My comments are as follows:

Major considerations:

1. In Fig1, authors showed that IRX2 is induced in CFs by AngII treatment in vitro and in vivo. Is the ratio of IRX2+ CFs altered by AngII treatment?
2. Again, the characteristics of IRX2-positive CFs in vivo are rather superficial. Quantitative evaluation of IRX2+ CFs in heart failure is needed; are they comparable in TAC and AngII? Furthermore, do IRX2+ CFs continue to increase over time with these stimuli and remain present in the chronic phase?
3. Inhibition of the IRX2-Egr1 pathway in CF improves cardiac hypertrophy, and cardiac systolic function despite sustained stimulation with Ang II and TAC. These results suggest that the IRX2-Egr1 pathway may be a new therapeutic target for heart failure. Please clarify the mechanism of these improvement.
4. FigS2H-J is missing from the manuscript.
5. In Fig. 3, the authors demonstrated that MF-specific IRX2 depletion attenuates AngII or pressure overload-induced cardiac fibrosis. Please indicate whether IRX2 KO reduces the number of Postn+ MFs or alters the nature of Postn+ MFs. Does KO of IRX2, which is not expressed under normal conditions, cause CFs to revert from MFs to steady state CFs?
6. Fig. 4, are cardiac fibrosis and systolic dysfunction promoted by MF-specific IRX2 overexpression further exacerbated after 12 weeks compared to controls?
7. Again, please indicate whether IRX2 OE increases the number of Postn+ MFs?
8. In order to clarify whether the IRX2-Egr1 pathway functions as a fibrosis regulator, it is important to show whether it is a fibrosis mechanism independent of the already established fibrosis regulators, Sox9 and Meox1.
9. The authors' strategy cannot distinguish whether inhibition of the IRX2-Egr1 pathway in CFs prevents Ang II and pressure overload-induced cardiac fibrosis and cardiac dysfunction or restores these adverse changes. It would be a major advance if inhibition of this new pathway could restore already established cardiac fibrosis and contractile function.

MS # NCOMMS-22-46574

Title: IRX2 regulates angiotensin II-induced cardiac fibrosis by transcriptionally activating Egr1 in mice

Response to Reviewer #1

We thank the reviewer for the constructive suggestions very much

1. The rationale for choosing the IRX family to investigate in the context of cardiac fibroblast activation and fibrosis is not clear. In other words, it's a random selection rather than based on a big data analysis or a screening experiment. Addressing this will significantly improve the rationale/justification.

Re: Thanks for your important question.

We selected the IRX family to investigate in the context of cardiac fibroblast activation and fibrosis based on the previous scientific observations. **IRX family was not a random selection.** IRX family can be regulated by TGF- β , FGFs and SOX9, which are master regulators of cardiac fibrosis [1-4]. IRX2 was upregulated in a mouse model of DCM [5] and could be activated by mitogen-activated protein kinases [6-7]. We listed several scientific evidences in the table. We also rewrote this part to improve our rationale.

Authors (year)	Scientific findings	Reference
Díaz-Hernández ME, et al (2013)	Irx1 and Irx2 are coordinately expressed and regulated by retinoic acid, TGF β and FGF signaling	[1]
Scharf GM, et al (2019)	IRX2 has been shown to be regulated by SOX9	[2]
Martorell Ò, et al (2014)	Iro/IRX transcription factors negatively regulate Dpp/TGF- β pathway activity during intestinal tumorigenesis	[3]
Gómez-Skarmeta JL, et al (2002)	Xiro (the Xenopus analogue of a human IRX member) functions as a repressor of BMP-4 after activation by the Wnt signalling pathway	[4]
Cho et al (2019)	IRX2 was upregulated in a mouse model of	[5]

	DCM	
Matsumoto K(2004)	The transcription factor Irx2 was a target of FGF8/MAP kinase cascade.	[6]
Yao (2021)	JNK1/2 phosphorylated IRX3, leading to its dimerization and nuclear translocation for transcription	[7]
Bruneau BG (2001)	Irx4 KO mice develop spontaneous cardiomyopathy with impaired systolic function.	[8]

Reference

- [1] Díaz-Hernández ME, Bustamante M, Galván-Hernández CI, et al. Irx1 and Irx2 are coordinately expressed and regulated by retinoic acid, TGF β and FGF signaling during chick hindlimb development. *PLoS One*. 2013; 8(3):e58549.
- [2] Scharf GM, Kilian K, Cordero J, et al. Inactivation of Sox9 in fibroblasts reduces cardiac fibrosis and inflammation. *JCI Insight*. 2019;5(15):e126721.
- [3] Martorell Ò, Barriga FM, Merlos-Suárez A, et al. Iro/IRX transcription factors negatively regulate Dpp/TGF- β pathway activity during intestinal tumorigenesis. *EMBO Rep*. 2014; 15(11):1210-8.
- [4] Gómez-Skarmeta JL, Modolell J. Iroquois genes: genomic organization and function in vertebrate neural development. *Curr Opin Genet Dev*. 2002;12(4):403-8.
- [5] Cho E, Kang H, Kang DK, Lee Y. Myocardial-specific ablation of Jumonji and AT-rich interaction domain-containing 2 (Jarid2) leads to dilated cardiomyopathy in mice. *J Biol Chem*. 2019;294(13):4981-4996.
- [6] Matsumoto K, Nishihara S, Kamimura M, et al. The prepattern transcription factor Irx2, a target of the FGF8/MAP kinase cascade, is involved in cerebellum formation. *Nat Neurosci*. 2004;7(6):605-12.
- [7] Yao J, Wu D, Zhang C, et al. Macrophage IRX3 promotes diet-induced obesity and metabolic inflammation. *Nat Immunol*. 2021;22(10):1268-1279.
- [8] Bruneau BG, Bao ZZ, Fatkin D, et al. Cardiomyopathy in irx4-deficient mice is preceded by abnormal ventricular gene expression. *Mol Cell Biol*. 2001; 21:1730–1736.

2. A recent paper, "Int J Cardiol . 2023 Jan 15;371:332-344 (PMID: 36181956) has showed, "IRX2 knockdown prevented mice from Ang II-induced ventricular dysfunction, cardiac hypertrophy, inflammation and fibrosis in mouse heart, and it decreased the levels of cardiac hypertrophy-related markers, oxidative stress response, and apoptosis of Ang II-treated cardiomyocytes." Of note, both papers employ the Ang II infusion as a primary model of cardiac stress. Therefore, this recently published paper needs to be cited, and the finding of the current paper need to be discussed with the reported results. However, this reviewer feels that the present paper is much more advanced with two FB-specific models. Thus, there is no concern about a lack of novelty/overlap, but it needs to be discussed.

Re: Thanks for your suggestion. We have cited this article (reference 43) and discussed (Page 14 line 18-21).

‘Surprisingly, but not unexpectedly, this attenuated fibrotic response was not observed in mice with cardiomyocyte-specific IRX2 depletion, suggesting that IRX2 derived from CFs, not cardiomyocytes, plays a critical role in Ang II-induced fibrotic remodelling in mice. Most importantly, IRX2 mfTg mice demonstrated a pathological phenotype and deteriorated cardiac function upon Ang II infusion. Inconsistent with our findings, Wang and his colleagues observed that lentivirus injection-induced IRX2 knockdown prevented mice from Ang II-induced cardiac hypertrophy in mouse heart. These incompatible results may be explained by the different manipulation of myocardial IRX2.’

3. Col1a2-Cre driven FB-targeting is questionable (PMID: 36289614, 34572061, 28436487). However, this concern is somewhat addressed with the Postn-Cre model. Nevertheless, the limitation associated with the Col1a2-Cre model should be discussed/acknowledged. This reviewer requests the authors to consider PostnMCM, and TCF21MCM to target residential FBs and activated FBs, respectively. In the FB field, a consensus is developing that these two models are currently the best tool to target cardiac fibroblasts (PMID: 34572061, 27447449).

Re: Thanks for your suggestion.

This kind reviewer questioned the efficacy of Col1 α 2-Cre driven CF-targeting strategy. With Col1 α 2-Cre driven CF-targeting strategy, Li and his colleagues found that ATF3 protected against cardiac fibrosis by suppressing MAP2K3-p38 signaling in mice [1]. He found that Col1 α 2 expression in heart, liver, lung, kidney, spleen, and vessels. **High level Col1a2 expression was only observed in heart and vessel, low level Col1a2 expression was observed in kidney, and no significant Col1a2 expression was observed in other organs. In ATF3 cTg mice, ATF3 proteins were significantly elevated only in the heart and vessel, slightly elevated in the kidney.** With this Col1 α 2-Cre driven CF-targeting strategy, Hind Lal and his colleagues found that tamoxifen treatment led to an ~60% reduction of GSK-3 β protein in the cardiac fibroblasts from KO mice compared to littermate controls [2]. In our study, we found that mouse CFs freshly isolated from IRX2 cfKO mice demonstrated ~61% IRX2 protein loss compared to CFs isolated from IRX2^{fl/fl} littermates. These data suggested that Col1 α 2-driven gene IRX2 deletion is effective.

We also agreed the opinion of this kind reviewer that TCF21MCM-Cre driven CF-targeting strategy might be a better strategy. Currently, we didn't have TCF21MCM on hand. **The use of Col1 α 2-driven IRX2 deletion didn't compromise our main finding that IRX2 is a master regulator of pathological cardiac fibrosis. The limitation of Col1 α 2-driven IRX2 deletion was partly addressed with the use of Postn-Cre-mediated IRX2 deletion.** We have discussed this limitation (Page 14 line 21-25).

Reference

- [1] Li Y, Li Z, Zhang C, et al. Cardiac Fibroblast-Specific Activating Transcription Factor 3 Protects Against Heart Failure by Suppressing MAP2K3-p38 Signaling. *Circulation*. 2017; 135(21):2041-2057.
- [2] Lal H, Ahmad F, Zhou J, Yu JE, Vagnozzi RJ, Guo Y, Yu D, Tsai EJ, Woodgett J, Gao E, Force T. Cardiac fibroblast glycogen synthase kinase-3 β regulates ventricular remodeling and dysfunction in ischemic heart. *Circulation*. 2014;130(5):419-30.

4. Targeting all fibroblasts with the Col1a2-Cre model led to a 61 % deletion; however, the more restrictive deletion strategy with the Postn-Cre model (myofibroblast only) led to 73% deletion. Authors need to explain this and discuss it in the text.

Re: Thanks for your interesting question.

The different deletion between the fibrosis models might be explained by the different Cre activity [1].

To induce Cre recombinase expression in IRX2 cfKO mice, **adult male mice were intraperitoneally injected with tamoxifen (25 mg/kg per day) for five consecutive days.**

Injury-inducible conditional myofibroblast-specific IRX2 mfKO were bred by crossing the obtained IRX2fl/fl mice with Postn-Cre mice. **These mice were intraperitoneally injected with tamoxifen (25 mg/kg per day) for five consecutive days beginning from the first day of Ang II infusion and fed with a diet containing 400 mg/kg tamoxifen citrate (#130860, Envigo) from the first day of Ang II infusion until the time of sacrifice.**

Reference

[1] Aguado-Alvaro LP, Garitano N, Abizanda G, Larequi E, Prosper F, Pelacho B. Comparative Evaluation of Inducible Cre Mouse Models for Fibroblast Targeting in the Healthy and Infarcted Myocardium. *Biomedicines*. 2022;10(10):2350.

5. In the α MHC-Cre-driven CM-specific model, 55 % deletion is significantly less than the vast literature with this model. Please clarify/discuss.

Re: Thanks for your important question.

IRX2 mRNA levels were much higher in CFs than in cardiomyocytes and endothelial cells. α -MHC-Cre-driven CM-specific model worked better in deleting gene with strong expression in cardiomyocytes.

6. Mouse age, tamoxifen treatment, and deletion blots are crucial; authors need to bring the experimental design sketch, deletion blot, etc., for all the models in the main

figures.

Re: As suggested, we have added animal design sketches and moved these information to the main figures.

7. Authors may want to change the word "foetal" to "fetal" to be consistent with the North American writing style like the rest of the manuscript text.

Re: As suggested, we have revised.

8. In the text, authors cited fig 2F-H). However, figure 2 and its legend ends at 2G only; there is no H. Please correct.

Re: As suggested, we have revised.

9. Fig 6H, and Fig S9A, it seems there are two EGR1 bands. The author must authenticate the correct band with the respective control (complete KO control samples).

Re: As suggested, we have provided new bands. For Figure 6A, the bands were confirmed by KO.

For Figure S10A (in the previous version Figure S9A), the bands were repeated.

10. Similarly, in Fig. S4, the IRX2 blot shows two bands. This need to be addressed.

Re: As suggested, we have provided a new band (In the revised version Figure S5A).

The single bands were confirmed by cmKO.

11. Furthermore, authors are requested to submit the uncut western blot images for all figures as an online-only figures.

Re: As suggested, we have provided all uncut western blot images.

Response to Reviewer #2

The problems with the manuscript are minor when weighed against its strengths, but the impact of the work will be greatly increased by addressing them.

We thank the reviewer for the constructive suggestions very much

1. The rationale for studying the Iroquois homeobox family of proteins in the context of CFs and cardiac fibrosis is lacking. There is an abrupt jump from “these proteins are expressed in the heart, and many have known roles in development” to “we studied whether these proteins impact CFs in cardiac fibrosis.” The rationale for studying *Irx2* particularly in this context, however, is strong scattered throughout the manuscript:

a. *Irx2* is induced by *Fgf8* and *TGF-β*;

b. MAPK phosphorylation activates *Irx2* transcription activation and inhibits *Irx2* transcription repression;

c. *Irx2* is regulated by *Sox9*, a key regulator of cardiac fibrosis;

d. *Irx2* (and *Irx1*) were upregulated in a mouse model of *DC(1, 2)M*;

Moving or copying (b) (lines: 270-1) and (d) (lines: 95-6) to the introduction will start the results on firmer footing.

Re: Thanks for your great suggestion. We also rewrote this part to improve our rationale

(Page 3 line 25-29).

2. The manuscript and figures do an excellent job of selecting and expounding on Egr1 as a major mechanism through which Irx2 can regulate fibrosis, but a brief primer on what is known about Egr1 promoting fibrosis in the heart and other organs would be beneficial. Additionally, are the global Egr1 knockout mice protected against AngII-induced fibrosis? Reference (28) of this manuscript (PMID: 21511034), which is used in the rationale for Egr1 selection cites an article (PMID: 10710348) where global Egr1 knockout increased the number of fibrotic loci basally and in response to adrenoreceptor agonists. If this is also true for the Egr1 knockout mice used in this study, it does not invalidate the authors conclusions, but it does complicate the picture and should be addressed in the discussion.

Re: Thanks for your important suggestion.

We have provided background information about Egr1 in cardiac fibrosis. Please see result section (Page 10 line 25-28).

As mentioned by this kind reviewer, a study (PMID: 10710348) found that global Egr1 knockout increased the number of fibrotic loci basally and in response to adrenoreceptor agonists. We carefully read this article, the authors compared fibrotic loci between 3 groups (vehicle-treated Egr1^{+/+} mice, vehicle-treated Egr1^{-/-} mice, or Iso + PE-treated Egr1^{-/-} mice). They found that global Egr1 knockout increased the number of fibrotic loci basally. However, this article had some limitations. First, the authors used H&E staining to evaluate cardiac fibrosis, which was not an accurate approach to detect fibrotic area. Second, the author didn't provide the statistical result of cardiac fibrosis between these 3 groups. The data in our study also demonstrated the key role of Egr1 in cardiac fibrosis that Egr1 deficiency significantly decreased Ang II-induced cardiac fibrosis in IRX2 Tg^{fl/fl} littermates (Figure 8B-C).

As suggested, we have discussed this in the discussion section (Page 15 line 15-18).

3. Because the protein abundance of Irx2 does not impact fibrotic phenotypes basally, but only after profibrotic stimulus, there must be a post-translational upstream switch for activating Irx2 transcription activation. The authors heavily hint that this switch is phosphorylation by MAPK induced by Ang II or pressure, and I am inclined to accept that. The scope and novelty of this study is in its identification of the downstream mechanism of Irx2. However, the question of upstream activation arises immediately given that in Figure 1 AngII increases Irx2 expression, but increasing Irx2 expression in Figure 2 is not sufficient to basally increase fibrosis. The simplest way to address this would just be to explore alternative hypotheses for upstream regulation in the discussion, but experiments with CFs expressing phospho-mimetic and phospho-dead Irx2 mutants or something similar nodding to upstream regulation would add to the study's impact.

Re: As said, the scope and novelty of this study is in its identification of the downstream mechanism of IRX2. In this study, we did a lot of job for selecting Egr1 as a major mechanism through which IRX2 can regulate fibrosis. This was the main part of our study.

This reviewer suggested that we performed additional experiments with CFs expressing phospho-mimetic and phospho-dead IRX2 mutants. We didn't perform this experiment for several reasons. First, Matsumoto et al reported that Mek1 phosphorylated the IRX2 protein (Ser326) to activate IRX2 **in chick** ^[1]. Whether this finding worked well in mice remained verified. Second, there was no commercial phosphorylated IRX2 antibody (Ser326). Without this antibody, we have difficulty to detect the activation of IRX2 in

mice. Three, we can't solve all question about IRX2 in one article, that is quite a large story. In this study, we focused our attention on identification of the downstream mechanism of IRX2. We will decipher the post-translational upstream switch for activating IRX2 transcription activation in our further study. We also discussed this in the discussion section (Page 15 line 2-8).

Reference

[1] Matsumoto K, Nishihara S, Kamimura M, et al. The prepattern transcription factor *Irx2*, a target of the FGF8/MAP kinase cascade, is involved in cerebellum formation. *Nat Neurosci*. 2004 Jun;7(6):605-12.

4. The authors mutated the putative *Irx2* binding sites in the *Egr1* promoter in a luciferase assay and showed that this reduced *Irx2*-dependent luciferase activity. The most straightforward way to demonstrate that *Irx2* promotes fibroblast activation directly through interactions with the *Egr1* gene would be to either mutate those binding sites in isolated CFs or design a CRISPRd system to competitively inhibit *Irx2* at those specific *Egr1* binding sites and show a reduction in α -SMA and/or Postn expression and positive cells and ECM proteins after AngII treatment. As the manuscript stands, it is clear that *Irx2* can induce *Egr1* transcription and that loss of *Egr1* prevents *Irx2*-dependent fibrotic remodeling. While this circumstantially suggests that *Irx2* is regulating fibrosis by binding and activating transcription at the *Egr1* gene, the more direct experiment would show this conclusively, adding to the significance of the current study.

Re: As suggested, we performed new experiments.

To confirm that IRX2 promoted fibroblast activation directly through interaction with the *Egr1* promoter, IRX2-overexpressed CFs were infected with an adenovirus carrying mutated *Egr1* promoter with two IRX2 binding sites deletion. After these IRX2 binding sites were mutated, IRX2-mediated effect was abrogated, as reflected by α -SMA mRNA level (Figure 7E).

5. In Figure 2G, representative echocardiograms need to be included in the figure or at least in a corresponding supplementary figure.

Re: As suggested, we have provided representative echocardiograms (Supplementary Figure 3C).

*6. For Figures 5C and 7D (as well as for sections taken from hearts after saline or Ang II treatment), it will be meaningful to show the percentage of α SMA and/or Postn positive cells alongside the changes in fluorescence intensity. A global reduction in just the α SMA fluorescence intensity is not adding anything to the immunoblot showing the same thing and westerns are better indicators of protein abundance than IF micrographs. However, if there is a significant reduction in the number of α SMA (and especially Postn given it is the Cre driver) positive cells in the Ang II treated *Irx2* knockout cells, then that is highly suggestive of a defect in fibroblast to myofibroblast conversion rather than a defect in myofibroblast fibrotic activity.*

Re: Thanks for your suggestion. We have provided the statistical result of α -SMA-positive cells (Figures 5C, 7D, and supplementary Figure 10C).

*7. Is there a rationale for using the *Coll1a2-CreER* mouse over the better characterized *Tcf21-CreER* mouse? The authors include adequate controls to address this concern, and none of the conclusions hinge on the *Coll1a2-cre* mouse anyway because the authors immediately shift to the more physiologically relevant injury-inducible model.*

Re: Just as this kind reviewer said, injury-inducible model was of great clinical significance. With this injury-inducible model, we revealed that IRX2 played a pro-fibrotic role. We used this *Coll1a2-CreER* mouse for the reason that this mouse line was also commonly used to manipulate gene in fibroblasts^[1-2]. With *Coll1 α 2-Cre* driven CF-targeting strategy, Li and his colleagues found that ATF3 protected against cardiac fibrosis by suppressing MAP2K3-p38 signaling in mice^[1]. He found that *Coll1a2* expression in heart, liver, lung, kidney, spleen, and vessels. **High level *Coll1a2* expression was only observed in heart and vessel, low level *Coll1a2* expression was**

observed in kidney, and no significant Col1a2 expression was observed in other organs. In ATF3 cTg mice, ATF3 proteins were significantly elevated only in the heart and vessel, slightly elevated in the kidney. With this Col1 α 2-Cre driven CF-targeting strategy, Hind Lal and his colleagues found that tamoxifen treatment led to an ~60% reduction of GSK-3 β protein in the cardiac fibroblasts from KO mice compared to littermate controls [2]. In our study, we found that mouse CFs freshly isolated from IRX2 cfKO mice demonstrated ~61% IRX2 protein loss compared to CFs isolated from IRX2^{fl/fl} littermates. Col1 α 2-driven gene IRX2 deletion is effective. We also agreed the opinion that TCF21MCM-Cre driven CF-targeting strategy might be a better strategy. Currently, we didn't have TCF21MCM on hand. **The use of Col1 α 2-driven IRX2 deletion didn't compromise our main finding that IRX2 is a master regulator of pathological cardiac fibrosis. The limitation of Col1 α 2-driven IRX2 deletion was partly addressed with the use of Postn-Cre-mediated IRX2 deletion.** We have discussed this limitation (Page 14 line 21-25).

Reference

- [1] Li Y, Li Z, Zhang C, et al. Cardiac Fibroblast-Specific Activating Transcription Factor 3 Protects Against Heart Failure by Suppressing MAP2K3-p38 Signaling. *Circulation*. 2017; 135(21):2041-2057.
- [2] Lal H, Ahmad F, Zhou J, Yu JE, Vagnozzi RJ, Guo Y, Yu D, Tsai EJ, Woodgett J, Gao E, Force T. Cardiac fibroblast glycogen synthase kinase-3 β regulates ventricular remodeling and dysfunction in ischemic heart. *Circulation*. 2014;130(5):419-30.

8. Figure 6F could be made clearer by adding to the boxplot itself that it is a ChIP-PCR for the Egr1 promoter. The legend and text state this clearly, but having the plot clear that it is measuring Egr1 will be helpful to someone skimming the figures. Taking a broader view, it would help to clarify for each occurrence (ideally in the figure itself but at least in the legend) what each color is for all the IF micrographs.

Re: As suggested, we have revised.

9. Figure S7 would be clearer if the color legend for the padj values were removed for both A and B, adding a note in the legend that only highly significant pathways (whatever the cutoff is, be it $p_{adj} < 0.01$ or just 30 and 20 most significant, respectively) are shown.

Re: As suggested, we have revised (in the revised version, supplementary Figure 8).

Response to Reviewer #3

This manuscript elegantly demonstrates the role of Irx2 in cardiac fibrosis. However, to argue that Irx2 is the "master regulator of cardiac fibrosis", it would be desirable to clarify whether there is a same fibrotic mechanism mediated by the IRX2-Egr1 pathway in a more physiological model of cardiac fibrosis, myocardial infarction.

Re: We thank the reviewer for the constructive suggestions very much.

Myocardial infarction is a typical example of reparative fibrosis, as sudden death of a large number of cardiomyocytes stimulates inflammation and subsequent activation of reparative myofibroblasts, leading to formation of a scar ^[1]. The scar lacks contractile capacity, but serves a critical protective role, maintaining the structural integrity of the chamber, and preventing cardiac rupture. In other cardiac diseases, fibrosis predominantly involves the interstitium, and develops insidiously in the absence of significant cardiomyocyte loss. Systemic hypertension is associated with progressive interstitial and perivascular deposition of ECM proteins that increase myocardial stiffness ^[1]. **Hypertension-related cardiac fibrosis, which could be mimicked by TAC surgery or Ang II infusion, is also a physiological model of cardiac fibrosis.** In this article, we have used two hypertensive models to investigate the role of IRX2. We didn't establish myocardial infarction model to verify the role of IRX2 for the reason that **the findings obtained from hypertensive models can't be easily translocated into a myocardial infarction model.** Though we didn't confirm our findings in myocardial infarction model, our data also had great translational potential for the reason that hypertension patients were so huge, and there were no drugs that could prevent hypertension-related cardiac fibrosis.

Reference

[1] Frangogiannis NG. Cardiac fibrosis. *Cardiovasc Res.* 2021;117(6):1450-1488.

1. In Fig1, authors showed that IRX2 is induced in CFs by Ang II treatment in vitro and in vivo. Is the ratio of IRX2+ CFs altered by Ang II treatment?

2. Again, the characteristics of IRX2-positive CFs in vivo are rather superficial. Quantitative evaluation of IRX2+ CFs in heart failure is needed; are they comparable in TAC and Ang II? Furthermore, do IRX2+ CFs continue to increase over time with these stimuli and remain present in the chronic phase?

Re: Your question is quite important.

In our previous manuscript, we have provided the representative images of IRX2+ CFs (Figure S1C).

To end this question, we provided the statistical results of IRX2+ CFs in TAC or Ang II-treated mice. The data in our results suggested that TAC or Ang II significantly increased IRX2+ CFs. Furthermore, IRX2+ CFs continued to increase over time with Ang II infusion (from 4 weeks to 12 weeks).

3. Inhibition of the IRX2-Egr1 pathway in CF improves cardiac hypertrophy, and cardiac systolic function despite sustained stimulation with Ang II and TAC. These results suggest that the IRX2-Egr1 pathway may be a new therapeutic target for heart failure. Please clarify the mechanism of these improvement.

Re: As suggested, we have discussed the mechanism of these improvement (Page 16 line 3-9).

“In our study, we also found that myofibroblast-specific IRX2 depletion, but not cardiomyocyte-specific IRX2 depletion, could attenuate cardiac hypertrophy, and improve cardiac systolic function. Growth factors and cytokines produced by CFs, were closely involved into the development of cardiac hypertrophy via paracrine-mediated cell-cell communications. The decreased secretion of myofibroblasts-derived pro-hypertrophic factors (TGF- β , PDGF-A and PDGF-B) in mice with myofibroblast-specific IRX2 depletion in response to Ang II infusion might explain the improvement in cardiac hypertrophy and systolic function.”

4. FigS2H-J is missing from the manuscript.

Re: We apologized for this.

According to Figure legends, Fig S2H-J is β -myosin heavy chain (β -MHC) mRNA (H), systolic blood pressure (I) and heart rate (J). We have provided these data (Supplementary Figure 3B-E).

5. In Fig. 3, the authors demonstrated that MF-specific IRX2 depletion attenuates Ang II or pressure overload-induced cardiac fibrosis. Please indicate whether IRX2 KO reduces the number of Postn⁺ MFs or alters the nature of Postn⁺ MFs. Does KO of IRX2, which is not expressed under normal conditions, cause CFs to revert from MFs to steady state CFs?

Re: As suggested, we have performed new experiments. We found that myofibroblast-specific IRX2 depletion reduced the number of Postn⁺ myofibroblasts (Supplementary Figure 4C). We also found that IRX2 knockdown decreased Ang II-induced Col1 production in myofibroblasts (Figure 5E).

The second question is that whether KO of IRX2, which is not expressed under normal conditions, cause CFs to revert from myofibroblasts to steady state CFs. Using cells with IRX2 depletion under normal conditions, we didn't distinguish whether IRX2 deficiency prevented fibroblast-to-myofibroblast conversion or reverted from myofibroblasts to steady state CFs. **Actually, fibroblast-to-myofibroblast conversion and reversion from myofibroblasts to steady state CFs are two different**

pathological processes. The aim of our study was to decipher the role of CF activation and fibroblast-to-myofibroblast conversion. We never said that IRX2 depletion would cause myofibroblasts reprogrammed into CFs in our article.

In our study, we found that shRNA-mediated knockdown of IRX2 significantly suppressed the upregulation of α -SMA⁺ cells in response to Ang II treatment. IRX2 regulated TGF- β , PDGF-A, PDGF-B and PAI-1 expression in Ang II-infused CFs in an Egr1-dependent manner. TGF- β , PDGF-A and PDGF-B were important mediators initiating the process of fibrosis. Based on our data obtained, we can only conclude that IRX2 depletion attenuated the Ang II-induced transdifferentiation of fibroblasts into myofibroblasts. Though we didn't answer that whether KO of IRX2 could revert from myofibroblasts to steady state CFs, the main findings and main novelty of our study wouldn't be compromised for the reason that we have use several approaches to reveal the pro-fibrotic role of IRX2. The scope and novelty of this study is in its identification of the downstream mechanism of IRX2.

6. Fig. 4, are cardiac fibrosis and systolic dysfunction promoted by MF-specific IRX2 overexpression further exacerbated after 12 weeks compared to controls?

Re: Thanks for your question.

All mice in Figure 4 were killed at 4 weeks after Ang II infusion for the reason that at this time point, we have reached our research goal and found that myofibroblast-specific IRX2 gain of function promoted progressive cardiac fibrosis induced by Ang II and accelerated the deterioration of cardiac function in mice.

7. Again, please indicate whether IRX2 OE increases the number of Postn+ MFs?

Re: As suggested, we have performed new experiments. We found that myofibroblasts-specific IRX2 overexpression increased the number of Postn⁺ myofibroblasts (Supplementary Figure 6E).

8. In order to clarify whether the IRX2-Egr1 pathway functions as a fibrosis regulator, it is important to show whether it is a fibrosis mechanism independent of the already

established fibrosis regulators, Sox9 and Meox1.

Re: As suggested, we have performed new experiments.

We found that the pro-fibrotic effect of IRX2 overexpression was independent of Sox9 and Meox1 (n=5).

9. The authors' strategy cannot distinguish whether inhibition of the IRX2-Egr1 pathway in CFs prevents Ang II and pressure overload-induced cardiac fibrosis and cardiac dysfunction or restores these adverse changes. It would be a major advance if inhibition of this new pathway could restore already established cardiac fibrosis and contractile function.

Re: Thanks for your important question.

The aim of our study was to decipher the role of CF activation and fibroblast-to-myofibroblast conversion. In our article, we didn't distinguish whether inhibition of the IRX2-Egr1 pathway prevented or reversed Ang II-induced cardiac fibrosis.

We found that IRX2 regulated TGF- β , PDGF-A, PDGF-B and PAI-1 expression in Ang II-infused CFs in an Egr1-dependent manner. TGF- β , PDGF-A and PDGF-B were important mediators initiating the process of fibrosis. In this aspect, we concluded that inhibition of the IRX2-Egr1 pathway might prevent Ang II-induced cardiac fibrosis. Though we didn't answer that whether IRX2 depletion could reverse the transdifferentiation of fibroblasts into myofibroblasts, the main findings and main novelty of our study wouldn't be compromised for the reason that we have used several approaches to reveal the pro-fibrotic role of IRX2. The scope and novelty of this study

is in its identification of the downstream mechanism of IRX2.

Your question quite inspired us to continue. To discover the specific inhibitor of IRX2-Egr1 pathway, we now focus on the post-translational upstream switch for activating IRX2 transcription activation. With a specific IRX2 inhibitor, we can easily decipher whether inhibition of the IRX2-Egr1 pathway reverses Ang II-induced cardiac fibrosis in mice.

REVIEWERS' COMMENTS

Reviewer #1 (Remarks to the Author):

Thanks much for taking care of my concerns and suggestions.

Reviewer #2 (Remarks to the Author):

The authors have satisfactorily addressed all my concerns and questions. The revision is thorough and comprehensive, thus has improved the quality of manuscript. The current version is acceptable for publication.

Reviewer #3 (Remarks to the Author):

Ma and colleagues have substantially revised their manuscript and provide compelling evidence that the fibrotic mechanism is mediated by the IRX2-Egr1 pathway in two hypertensive models. However, it seems insufficient to conclude that IRX2 is a master regulator of cardiac fibrosis, as they claim. Figures S1E,F clearly show that a certain number of IRX2+ CFs are present even at steady state, and in addition, Figure S4C shows that half of the Postn+ cells are retained even in the MF-specific IRX2 KO.

It may be possible to demonstrate the contribution of IRX2 to cardiac fibrosis by showing the percentage of IRX2+CF and Postn+CF to total CFs in the newly added Figures S1E, F, S4C and S6E. The quantification method should be clarified.

For Figures S4C and S6E, it would be desirable to have the DAPI merge figures.

MS # NCOMMS-22-46574A

Title: IRX2 regulates angiotensin II-induced cardiac fibrosis by transcriptionally activating Egr1 in mice

Response to Reviewer #1

1. Thanks much for taking care of my concerns and suggestions.

We thank the reviewer for the constructive suggestions very much.

Response to Reviewer #2

1. The authors have satisfactorily addressed all my concerns and questions. The revision is thorough and comprehensive, thus has improved the quality of manuscript. The current version is acceptable for publication.

We thank the reviewer for the constructive suggestions very much.

Response to Reviewer #3

1. Figures S1E,F clearly show that a certain number of IRX2+ CFs are present even at steady state, and in addition, Figure S4C shows that half of the Postn+ cells are retained even in the MF-specific IRX2 KO.

It may be possible to demonstrate the contribution of IRX2 to cardiac fibrosis by showing the percentage of IRX2+CF and Postn+CF to total CFs in the newly added Figures S1E, F, S4C and S6E. The quantification method should be clarified. For Figures S4C and S6E, it would be desirable to have the DAPI merge figures.

Re: To end this question, we detected the percentage of POSTN+/IRX2+ CFs. We also used DAPI to label nuclei. Please see Figure S1E-G, Figures S4C and Figure S6E. We have clarified quantification method.